# Exploring the impact of antibody-dependent cellular phagocytosis-related genes on the prognosis of metastatic melanoma

**Junhao Chen[1]❀, Jiapeng He[2]❀, Xiaolong Xu[3]❀, Haiyan Sun[4]\*, Jianglin Zhang🄶[1]\***

**1** Department of Dermatology, Shenzhen People's Hospital, The Second Clinical Medical College, Jinan University, Shenzhen, Guangdong, China, **2** Department of Biochemistry, School of Medicine, Southern University of Science and Technology, Shenzhen, Guangdong, China, **3** Department of Plastic Surgery, Xiangya Hospital, Central South University, Changsha, China, **4** Department of Experimental Research, South China Hospital, Medical School, Shenzhen University, Shenzhen, P. R. China

❀ These authors contributed equally to this work.
\* 18302099013@163.com (HS); leozjl1010@126.com (JZ)

## Abstract

### Background

Metastatic melanoma is a challenging clinical condition with poor prognosis. Recent research has highlighted the role of antibody-dependent cellular phagocytosis (ADCP) in tumor immunity, suggesting prognostic implications for ADCP-related genes (ARGs). This study develops a prognostic model for metastatic melanoma using ARGs to enhance clinical decision-making and therapeutic strategies.

### Methods

Prognostic ARGs were identified from the GSE46517 and GSE7553 datasets. A prognostic model was constructed using LASSO-Cox regression and validated across multiple cohorts, including TCGA and GEO datasets. A nomogram was developed to assess survival outcomes in metastatic melanoma patients. Functional assays, including siRNA knockdown of DOCK10 in A375 cells, were conducted to validate the role of DOCK10 in melanoma progression.

### Results

A prognostic model based on six ARGs—NDRG1, HRAS, KPNA2, ICAM1, DOCK10, and CDC20—was developed. Patients were stratified into high- and low-risk groups based on risk scores, with high-risk patients showing poorer overall survival (OS) in both validation cohorts. The model was validated as an independent prognostic factor. Gene set enrichment analysis (GSEA) indicated that the low-risk group was enriched in immune-related pathways. High-risk patients exhibited higher genomic

**Data availability statement:** The data-sets generated and/or analyzed during the current study are available in the figshare repository at https://figshare.com/articles/dataset/_b_Exploring_the_impact_of_Antibody-Dependent_Cellular_Phagocytosis-Related_Genes_on_the_prognosis_of_meta-static_melanoma_b_Item/29143184. For data access requests, researchers may contact the Research Data Management Office at Southern University of Science and Technology at: 12331434@mail.sustech.edu.cn. This office serves as a non-author point of contact and will handle all data-related inquiries independent-ly of the research team. All other datasets analyzed in this study are publicly available from The Cancer Genome Atlas (TCGA-SKCM) at https://portal.gdc.cancer.gov/ and Gene Expression Omnibus (GEO) database at http://www.ncbi.nlm.nih.gov/geo.

**Funding:** National Natural Science Foundation of China (82073018 and 82073019), Shenzhen Science and Technology Innovation Commission, China (Shenzhen Natural Science Foundation, JCYJ20210324114212035),The State Key Program of National Natural Science of China(Grant No.62331011 ).

**Competing interests:** The authors declare that they have no competing interests.

instability, which was associated with poorer prognosis. Knockdown of DOCK10 in A375 cells significantly reduced proliferation, migration, and invasion, confirming its role in melanoma progression.

## Conclusion

The model also demonstrated associations with immune cell infiltration and drug sensitivity, highlighting its potential utility in optimizing immunotherapy and chemo-therapy strategies. This study developed a novel ARG-based prognostic model that aids in survival prediction and therapeutic decision-making for metastatic melanoma patients. DOCK10 was identified as a potential therapeutic target in melanoma metastasis.

## Introduction

Melanoma is a malignant tumor originating from melanocytes and can occur in the skin, eyes, meninges, and various mucosal surfaces [1]. Although cutaneous mela-noma accounts for only 2% of skin cancers, it is responsible for 90% of skin tumor-related deaths [1–3]. Cutaneous melanoma is characterized by its insidious onset, high invasiveness, and strong metastatic potential, with a tumor thickness of just 4 mm indicating a high risk of cancer spread and mortality [4,5]. While early-stage melanoma can be managed with surgical resection for a favorable prognosis, met-astatic melanoma is one of the deadliest stages, leading to poor prognosis and low survival rates for patients [6,7]. Therefore, developing new effective prognostic prediction models is needed to improve the clinical management of metastatic mela-noma and provide potential targets for clinical treatment.

In recent years, targeted therapies and immunotherapies have been exten-sively employed in metastatic melanoma individuals, showing great potential [8]. Immune checkpoint inhibitors such as anti-PD-L1 (programmed cell death ligand 1) and anti-CTLA-4 (cytotoxic T lymphocyte antigen 4) have successfully reduced cancer mortality rates in patients [9,10]. However, the effectiveness of immuno-therapy varies significantly among individuals, with nearly 50% of patients show-ing no response or developing resistance to these handlings [8,11]. Therefore, further exploration of immune regulatory mechanisms within the tumor microenvi-ronment of melanoma is of great implication for predicting metastatic melanoma prognosis.

ADCP (Antibody-Dependent Cellular Phagocytosis) is an antibody-arbitrated immune effector mechanism that has shown to possess therapeutic impact in tumor [12,13]. Specifically, macrophages recognize the fragment crystallizable (Fc) portion of antibodies through Fcγ receptors (FcγR), activating phagocytosis to eliminate antibody-tagged tumor cells [14]. Besides directly clearing tumor cells, ADCP can modulate the cancer microenvironment by persuading anti-tumor immune responses or inhibiting angiogenesis and the tumor-favorable microenvironment [15–17]. Although studies have shown that blocking the CD47-SIRPα interaction can enhance

the anti-tumor effect of immunotherapy in mouse melanoma models through ADCP, the function of ADCP-related genes (ARGs) in metastatic melanoma has not been extensively deliberate. Strong information regarding the effects of ARGs on the prognosis, tumour immunological milieu, and clinical management of metastatic melanoma is lacking.

In order to predict the prognosis for individuals with metastatic melanoma, the present research employed bioinformatics analysis to create an ARGs-based risk score using data from the Gene Expression Omnibus (GEO) and The Cancer Genome Atlas (TCGA). Furthermore, the relationship between the risk score and immune cell infiltration was examined, as was its capacity to forecast medication sensitivity and the effectiveness of immunotherapy. To strengthen the biological relevance of our findings, we performed comprehensive functional validation of DOCK10, a key gene in our prognostic model, demonstrating its critical role in melanoma cell proliferation, migration, and invasion. The ARG-based risk score offers a theoretical foundation for more individualised approaches to therapy and may be used as a prognostic indicator and therapeutic target for metastatic melanoma.

## Materials and methods

### Data acquisition for melanoma patients

The data used in this study were obtained from TCGA (The Cancer Genome Atlas) database (https://portal.gdc. cancer.gov/) under the project identifier TCGA-SKCM and GEO (Gene Expression Omnibus) database (http://www. ncbi.nlm.nih.gov/geo). A cohort of 351 metastatic melanoma samples, including data of gene expression and clinical information, was retrieved from the TCGA dataset. GSE7553 dataset contains 14 primary melanoma samples and 40 metastatic melanoma samples, while the GSE46517 dataset includes 31 primary and 73 metastatic tissue samples. The GSE7553 and GSE46517 datasets were used to identify differentially expressed genes (DEGs) among primary and metastatic samples. Additionally, GSE19234, GSE54467, and GSE65904 datasets, comprising 38, 79, and 210 metastatic melanoma samples, respectively, were used as validation cohorts. The "limma" program in R software was used to screen the DEGs, and the criteria were set to $|log2FC| > 1$ and adjusted $p < 0.05$ [18]. For TCGA cohorts, quantile normalization was conducted via the 'limma' package in R software. GEO datasets underwent ComBat correction using the sva package for batch effect removal. All expression values were log2-transformed pre-analysis for normality assurance.

### Identification of ADCP-related genes

ADCP-related genes were obtained from a study by Roarke A. Kamber et al. [19]. In this research, a genome-wide CRISPR knockout and overexpression screen was used in cancer cells and macrophages to systematically identify factors that impede ADCP without bias. A total of 543 ADCP-related genes with $p < 0.05$ were identified. These genes are involved in various cellular processes including cell surface antigen presentation, Fc receptor signaling, phagocytic machinery, and cytoskeletal reorganization.

To identify ADCP-related genes with potential prognostic significance in metastatic melanoma, we performed univariate Cox regression analysis on the TCGA cohort and identified 181 genes significantly associated with survival ($p < 0.05$). By overlapping these survival-associated genes with 543 ADCP-related genes, we identified nine ADCP-related genes with prognostic significance (ARGs): NDRG1, HRAS, KPNA2, CCNB1, ICAM1, KIF4A, DOCK10, CDC20, and BLM.

### Construction of the prognostic risk scoring model

For our initial analysis, a training set and a validation set were randomly assigned to the TCGA cohort in a 6:4 proportion. To identify potential prognostic risk genes for metastatic melanoma in the TCGA training set, univariate Cox regression analysis was performed using overall survival (OS) data, which included survival time (in months) and survival status (alive/dead) as the target variables. Genes with $p < 0.05$ were selected for further analysis. The potential prognostic

genes were then overlapped with ARGs to identify ARGs associated with metastatic melanoma prognosis. The "glmnet" package in R software was used to conduct the least absolute shrinkage and selection operator (LASSO) regression analysis in order to further refine the candidate genes. Finally, multivariate Cox regression analysis was used to build a prognostic risk score model for people with metastatic melanoma. The risk score was not a pre-existing variable in the dataset but was calculated based on the multivariate Cox regression coefficients and gene expression values. The following algorithm was used to determine each patient's risk score for metastatic melanoma based on their gene expression profiles:

Risk score $= \Sigma ni = \beta i \times Ei$ where $Ei$ represented the gene expression value, and $\beta i$ was the regression coefficient.

An additional analysis using the entire TCGA dataset (n = 351) was performed for model development to strengthen the statistical power for coefficient estimation. The model coefficients remained stable with both approaches, confirming the robustness of our six-gene signature.

## Performance evaluation and validation of the prognostic risk scoring model

Based on established practices in cancer prognostic studies [20,21], patients in the TCGA training set, TCGA validation set, and GSE19234, GSE54467, and GSE65904 validation cohorts were categorised as high-risk or low-risk using the median risk score calculated from the TCGA training set as the threshold. This approach ensures balanced group sizes and has been widely applied in similar prognostic model studies. Survival disparities across the various risk categories were compared using log-rank testing and Kaplan-Meier curves. ROC curves were created using the R software's "timeROC" package to evaluate the risk scoring model's prediction power. The viability of the survival model was further assessed using meta-analysis. To examine the distribution of risk scores across categories based on different clinico-pathological factors, such as age, gender, and stage, box plots were created using the Kruskal-Wallis test. Additionally, prognostic variations among high-risk and low-risk categories were evaluated in several subcategories stratified by clinical stage (I, II, III, IV), gender (male or female), and age (<60 or ≥60).

## Clinical characteristics based nomogram construction

In order to confirm if the risk score is an independent risk factor for metastatic melanoma, both univariate and multivariate Cox regression studies were conducted. Using the "rms" and "survival" packages in R software, a nomogram combining prognostic features was created based on the risk score and pertinent clinical parameters in order to predict the overall survival of individuals with metastatic melanoma for one, two, and three years. Plotting calibration curves was another way to evaluate the prediction model's accuracy and dependability.

## Enrichment analysis of differentially expressed genes in high- and low-risk categories

To investigate the biological roles linked to differentially expressed genes across high-risk and low-risk categories, Gene Ontology (GO) analysis was conducted using the "ClusterProfiler" tool in R software [22]. In Fig 5, the 'limma' package was employed with |log2FC|>1 and adjusted $p<0.05$ as thresholds to identify differentially expressed genes between high-risk and low-risk groups. Genes showing higher expression in the high-risk group (positive log2FC) were deemed 'up-regulated in high-risk' and linked to a worse prognosis. Conversely, those with higher expression in the low-risk group (negative log2FC) were regarded as 'down-regulated in high-risk' and associated with a better prognosis. The "GSVA" package in R was then used to perform gene set variation analysis (GSVA) in order to find enriched pathways in the low-risk and high-risk groups [23]. For GSEA analysis, the normalized enrichment score (NES) indicates the direction of pathway enrichment. Positive NES values represent pathways enriched in the high-risk phenotype, while negative NES values represent pathways enriched in the low-risk phenotype. GSVA pathways with a $p$-value<0.05 were considered significantly enriched [24].

## Immune cell infiltration analysis

CIBERSORT is a deconvolution algorithm based on gene expression matrices. To further evaluate immune microenvironment differences among the two categories, we employed CIBERSORT algorithm to estimate infiltration of 22 immune cell types in metastatic melanoma samples [25]. Additionally, the "TMEscore" was employed to investigate modifications in immunogenicity indicators among the high-risk and low-risk categories [26,27].

## Prediction of immunotherapy response

To measure potential impact of risk score on immunotherapy rejoinder, we analyzed PFS (progression-free survival) of high-risk and low-risk patients using RNA-seq data from 62 metastatic melanoma patients gained from the investigation by He et al [28]. We compared the number of responders (R) and non-responders (NR) among the two risk categories.

## Drug sensitivity analysis

The GSDC (Cancer Drug Sensitivity Genomics) database (https://www.cancerrxgene.org/) was employed to examine molecular characteristics associated with drug sensitivity/resistance [29]. "oncoPredict" package was employed to analyze tendency of different risk categories to respond to various chemotherapeutic agents [30]. This analysis predicted drug sensitivity in metastatic melanoma patients based on IC50 values (half maximal inhibitory concentration).

**scRNA-seq data analysis:** Single-cell RNA sequencing data from nine human melanoma samples (5 primary and 4 metastatic) were obtained from the GSE189889 dataset. Data preprocessing was performed using the Seurat package (v4.0.0) in R. Low-quality cells (<200 genes or >10% mitochondrial reads) were filtered out. Data normalization, scaling, and dimensionality reduction were performed using standard Seurat workflows. Cell clustering was performed using the shared nearest neighbor (SNN) modularity optimization-based clustering algorithm. Differential gene expression analysis between clusters was performed using the Wilcoxon rank-sum test.

## Experimental validation

**Cell culture and siRNA transfection:** A375 melanoma cells were cultured in DMEM supplemented with 10% fetal bovine serum and 1% penicillin/streptomycin at 37°C with 5% $CO_2$. For DOCK10 knockdown, cells were transfected with DOCK10-specific siRNA (5'-GAUCGCAGCUACUACGAAU-3') or control siRNA using Lipofectamine 3000 (Invitrogen) according to the manufacturer's protocol.

**Quantitative real-time PCR (qRT-PCR):** Total RNA was extracted using TRIzol reagent (Invitrogen), and 1 μg RNA was reverse-transcribed using the PrimeScript RT reagent kit (Takara). qPCR was performed using SYBR Premix Ex Taq (Takara) on a LightCycler 480 system (Roche). GAPDH served as the internal control. Primer sequences were as follows: DOCK10-F: 5'-ACGCAGAACCTGAAGAACCTG-3', DOCK10-R: 5'-GCCTCTTGCGGATCTGATAGG-3'; Ki67-F: 5'-ACGCCTGGTTACTATCAAAAGG-3', Ki67-R: 5'-CAGACCCATTTACTTGTGTTGGA-3'; CDH5-F: 5'-TTGGAACCAGATGCACATTGAT-3', CDH5-R: 5'-TCTTGCGACTCACGCTTGAC-3'; GAPDH-F: 5'-GGAGCGAGATCCCTCCAAAAT-3', GAPDH-R: 5'-GGCTGTTGTCATACTTCTCATGG-3'.

**Cell adhesion, migration and invasion assays:** 2 × 10^4 Human umbilical vein endothelial cells (HUVECs) expressing RFP (red fluorescent protein) are seeded in a 24-well plate. After the cells reach confluence, the culture medium is removed, and the cells are washed with PBS. Then, Calcein AM-stained A375 cells are resuspended in serum-free medium and seeded onto the 24-well plate. After 12 hours, the medium is changed, and the number of green fluorescent cells is observed under a fluorescence microscope. For wound healing assays, cells were grown to confluence, scratched with a pipette tip, and imaged at 0 and 24 hours. Migration distance was measured using ImageJ software. For invasion assays, 2 × 10^4 cells in serum-free medium were seeded in the upper chamber of Matrigel-coated Transwell inserts (8μm

pore size, Corning). Medium containing 10% FBS was added to the lower chamber as a chemoattractant. After 24 hours, invaded cells were fixed, stained with crystal violet, and counted in five random fields.

**Western blotting:** Cells were lysed in RIPA buffer supplemented with protease inhibitors. Equal amounts of protein (30 μg) were separated by SDS-PAGE and transferred to PVDF membranes. Membranes were blocked with 5% non-fat milk and incubated with primary antibodies against Ki67 (1:1000, Cell Signaling, #9449), ITGB1 (1:1000, Abcam, ab179471), CDH5 (1:1000, Cell Signaling, #2500), and GAPDH (1:5000, Proteintech, 60004–1-Ig) overnight at 4°C. After washing, membranes were incubated with HRP-conjugated secondary antibodies and visualized using ECL reagent (Millipore).

## Statistical analysis

The Chi-square test was used to analyse categorical variables, while the Mann-Whitney U test was used to analyse continuous variables. The relationship between two continuous variables was assessed using Spearman's correlation test. The "meta" package was used for meta-analysis, while the "glmnet" program was used for LASSO analysis. The "survival" program was used for survival analysis, and the Cox proportional hazards model was used for univariate and multivariate analyses as well as the log-rank test for category comparisons. In R software (version 4.2.1), all statistical analyses were two-tailed, and $p$-values $< 0.05$ were deemed highly significant.

## Results

In this study, we implemented a multi-step analytical pipeline that combined large-scale bioinformatic screening with targeted experimental validation (S1 Fig). First, we identified differentially expressed genes (DEGs) by comparing primary and metastatic melanoma samples from the GSE46517 and GSE7553 cohorts. These DEGs were then cross-referenced with antibody-dependent cellular phagocytosis (ADCP)-related genes reported by Kamber et al. [19] to nominate candidates with prognostic potential. Univariate Cox proportional hazards analysis in the TCGA melanoma cohort revealed a subset of survival-associated genes, which were further refined via LASSO regression. A six-gene prognostic signature was subsequently established using multivariate Cox regression and validated across multiple independent datasets. To interrogate the underlying biology, we performed pathway enrichment and immune cell infiltration analyses. Finally, we confirmed the functional relevance of DOCK10 through siRNA-mediated knockdown in melanoma cell lines, demonstrating its role in promoting melanoma progression.

### Identification of ADCP-related prognostic genes in metastatic melanoma

First, we used a threshold of adjusted $p < 0.05$ and $|log2FC| > 1$ to recognize 1,655 and 2,761 genes related to metastatic melanoma in the GSE46517 and GSE7553 datasets, respectively. Integrating the DEG lists from GSE46517 and GSE7553 yielded 887 overlapping genes associated with melanoma metastasis (Figs 1, S2 and S1 Table).These shared DEGs likely underpin the transition from primary tumors to metastatic lesions, we highlighted the top differentially expressed candidates enriched in key biological processes: cell cycle regulation (e.g., CCNB1, CDC20, CDK1), extracellular matrix remodeling (e.g., COL1A1, MMP1, FN1), and immune response modulation (e.g., CXCL8, IL6, CD274). This focused presentation emphasizes the coordinated reprogramming of proliferative, structural, and immunomodulatory pathways during melanoma metastasis.

Next, we performed univariate Cox regression analysis and identified 181 genes meaningfully related with survival ($p < 0.05$). By overlapping these genes with ADCP-related genes, we identified nine prognostic ADCP-related genes (ARGs): NDRG1, HRAS, KPNA2, CCNB1, ICAM1, KIF4A, DOCK10, CDC20, and BLM.

To further refine the list of prognostic genes, we conducted Lasso regression analysis on the nine selected ARGs. This approach helped us narrow down the key genes, ultimately identifying six ARGs for the construction of a prognostic model: NDRG1, HRAS, KPNA2, ICAM1, DOCK10, and CDC20. Heatmaps in Fig 1 illustrate the expression profiles

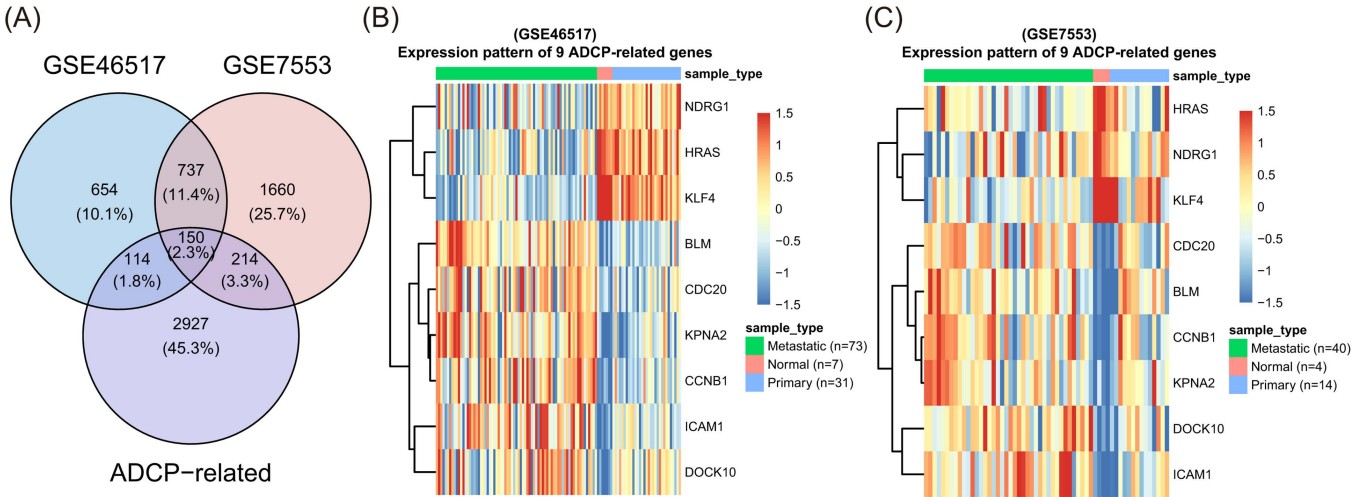

**Fig 1. Screening of ADCP-Related Differentially Expressed Genes Based on GSE46517 and GSE7553 Datasets. (A) illustrates the identification process of ADCP-related prognostic genes.** The Venn diagram shows the overlap between 887 differentially expressed genes (DEGs) from the GSE46517 and GSE7553 datasets and 543 ADCP-related genes from Kamber et al. (2021). This intersection yielded 9 genes (NDRG1, HRAS, KPNA2, CCNB1, ICAM1, KIF4A, DOCK10, CDC20, and BLM) that were both differentially expressed in metastatic melanoma and involved in ADCP mechanisms. Heatmaps in Fig 1B, C show the expression profiles of these nine ADCP-related genes across primary tumors, metastatic tumors, and normal tissues in the GSE46517 and GSE7553 datasets, respectively.

of these nine ADCP-related genes across primary tumors, metastatic tumors, and normal tissues, using data from the GSE46517 and GSE7553 datasets, respectively.

Among the six signature genes, DOCK10 was selected for in-depth functional validation. DOCK10, a member of the dedicator of cytokinesis (DOCK) family that orchestrates cytoskeletal remodeling and cell morphology which are key steps in metastatic dissemination with significant prognostic value (multivariate Cox coefficient = –0.13036; p = 0.003). Notably, DOCK10 expression was markedly higher in metastatic versus primary melanoma (fold change = 2.10; p < 0.001), underscoring its putative role in melanoma progression and justifying further experimental characterization.To delineate each signature gene's impact, we examined their expression dynamics and clinical correlations across cohorts. NDRG1 was significantly downregulated in metastatic relative to primary melanoma (fold change = 0.65; p < 0.001) and inversely associated with tumor stage (Spearman's r = –0.31; p < 0.001). HRAS and KPNA2 were upregulated in metastatic lesions (fold changes = 1.8 and 2.3, respectively; both p < 0.001) and positively correlated with proliferation markers. ICAM1 expression varied across samples but exhibited a strong positive correlation with immune cell infiltration scores (r = 0.58; p < 0.001). CDC20 was overexpressed in high-grade tumors and consistently associated with poorer overall survival. Although each gene individually predicted prognosis (all log-rank p < 0.05), the combined six-gene signature achieved superior discriminative power (AUC = 0.78) compared to any single marker (AUCs 0.61–0.71) (S3 Fig).

### Construction and validation of the prognostic risk scoring model for metastatic melanoma

Final selection of six prognostic ARGs was castoff to concept a prognostic model through multivariate Cox regression analysis. Resulting risk scoring formula was as trails:

Risk Score = - 0.11826 × Expr_NDRG1 + 0.12952 × Expr_HRAS + 0.22421 × Expr_KPNA2 - 0.13578 × Expr_ICAM1 - 0.13036 × Expr_DOCK10 + 0.05932 × Expr_CDC20

The TCGA cohort was split 6:4 across a training and validation set at random. The median cut-off value in the training set was used to divide the patients into high-risk and low-risk categories. Overall survival (OS) was considerably worse in

the high-risk population than in the low-risk category in the TCGA Training set, according to Kaplan-Meier (K-M) analysis (Fig 2A).

We assessed the risk score model's performance in the TCGA Validation set and external validation datasets GSE19234, GSE54467, and GSE65904 in order to further confirm its stability and generalisability. Patients were categorised into high-risk and low-risk groups using the same risk score methodology. Higher risk scores were linked to considerably poorer OS in the TCGA Validation, GSE19234 and GSE65904 (Fig 2B, C and E) populations, according to K-M analysis. While OS was lower in the high-risk group in the GSE54467 cohort, the distinction was not of statistical importance (Fig 2D). The OS was considerably shorter in the high-risk group than in the low-risk group (HR [95% CI]: 2.19 [1.74–2.75], $p < 0.01$), according to a meta-analysis that summarised the findings from all five cohorts (Fig 2F). According to our findings, the developed prognostic model offers metastatic melanoma sufferers useful prognostic information.

Risk scores were compared across patient subcategorys based on age, gender, and stage. The findings showed that, in comparison to female individuals, patients who were men had noticeably higher risk ratings (Fig 3). Risk ratings did not change significantly in statistical terms by stage or age. Additionally, survival analysis was carried out for high-risk and low-risk categories within various age, gender, and stage subcategories. Fig 3D demonstrates that the prognosis for the high-risk category was considerably worse than that of the low-risk group in a number of subcategories, especially those based on age and gender and individuals with stage II and III illness.

## Comprehensive evaluation of ADCP-based risk score: Clinical implications and nomogram

The impacts of gender, age, disease stage, and risk score on prognosis were examined using multivariate as well as univariate Cox regression analyses in order to further evaluate the independent predictive capacity of the ARG-based risk scoring model. According to univariate Cox analysis, the prognosis of patients with metastatic melanoma was correlated with gender, age, disease stage, and risk score. In individuals with metastatic melanoma, age (≥60 years), stage (III/IV), and high-risk score were found to be independent risk factors for a poor outcome in multivariate Cox analysis (Fig 4A). In order to improve the prognostic model's usefulness and clinical applicability, we created a nomogram that visualises the prognostic value of the ARG-based risk scoring model (Fig 4B) by combining clinical characteristics (age, stage) as well as the risk score to predict 1-, 2-, as well as 3-year OS of individuals with metastatic melanoma. Fig 4C's calibration curves demonstrated a high degree of agreement between the anticipated and actual outcomes.

## Biological function and pathway analysis related with the risk score

In order to investigate putative biological activities and mechanisms linked to the ARG-based risk score model, we conducted Gene Set Enrichment Analysis (GSEA) and compared GO annotations between the high-risk and low-risk categories. Skin-related processes such keratinisation, epidermis growth, skin growth, keratinocyte differentiation, and epidermal differentiation of cells were the primary areas of enrichment for genes with high expression in the high-risk category. Conversely, low-expression genes were predominantly associated with immune-related processes, such as the production of immunoglobulins, the production of immune response molecular mediators, the signaling pathway mediated by antigen receptors, the signaling pathway mediated by immune response-activating cell surface receptors, as well as the signaling pathway regulating immune response (Fig 5A). According to GSEA outcomes, the low-risk category was more likely to have processes linked to ferroptosis, the T cell receptor signaling pathway, as well as Th17 cell differentiation, whereas the high-risk category was more likely to have pathways linked to oxytocin, glucagon, oestrogen, and phospholipase D signaling (Fig 5B–D). According to these results, there may be benefits to using the ARG-based risk score model to comprehend the metabolic features and tumour microenvironment of metastatic melanoma.

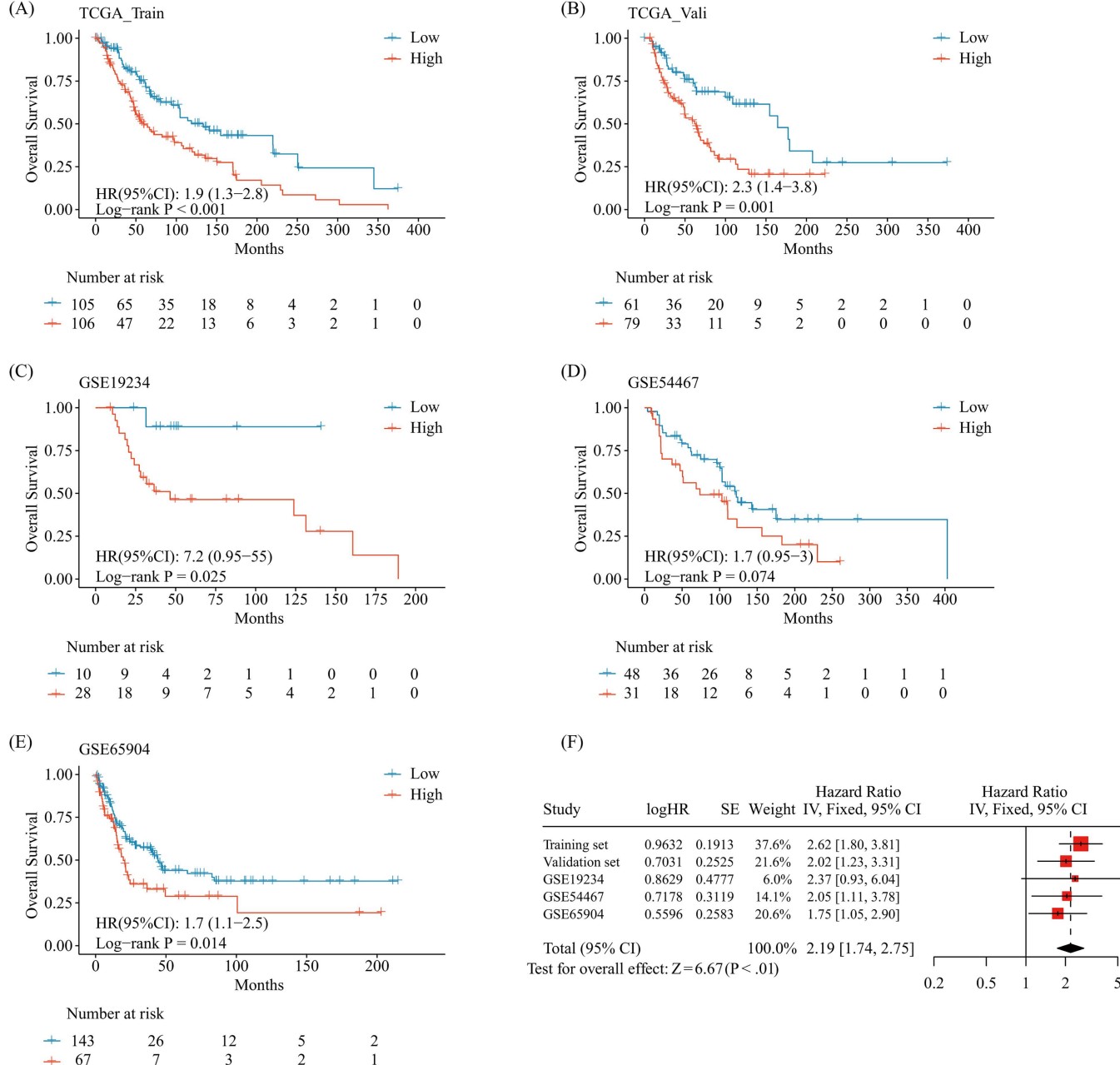

**Fig 2. Construction and validation of the prognostic model for metastatic melanoma based on ADCP-related genes.** (A) Kaplan-Meier (K-M) analysis of overall survival (OS) between high-risk and low-risk groups in the TCGA-Training cohort; (B-E) K-M analysis of OS between high-risk and low-risk groups in the TCGA-Validation cohort, GSE19234, GSE54467, and GSE65904 datasets; (F) Forest plot of the meta-analysis for prognostic comparison between high-risk and low-risk groups across the TCGA-Training, TCGA-Validation, GSE19234, GSE54467, and GSE65904 cohorts. ADCP, Antibody-Dependent Cellular Phagocytosis; K-M, Kaplan-Meier; OS, Overall Survival.

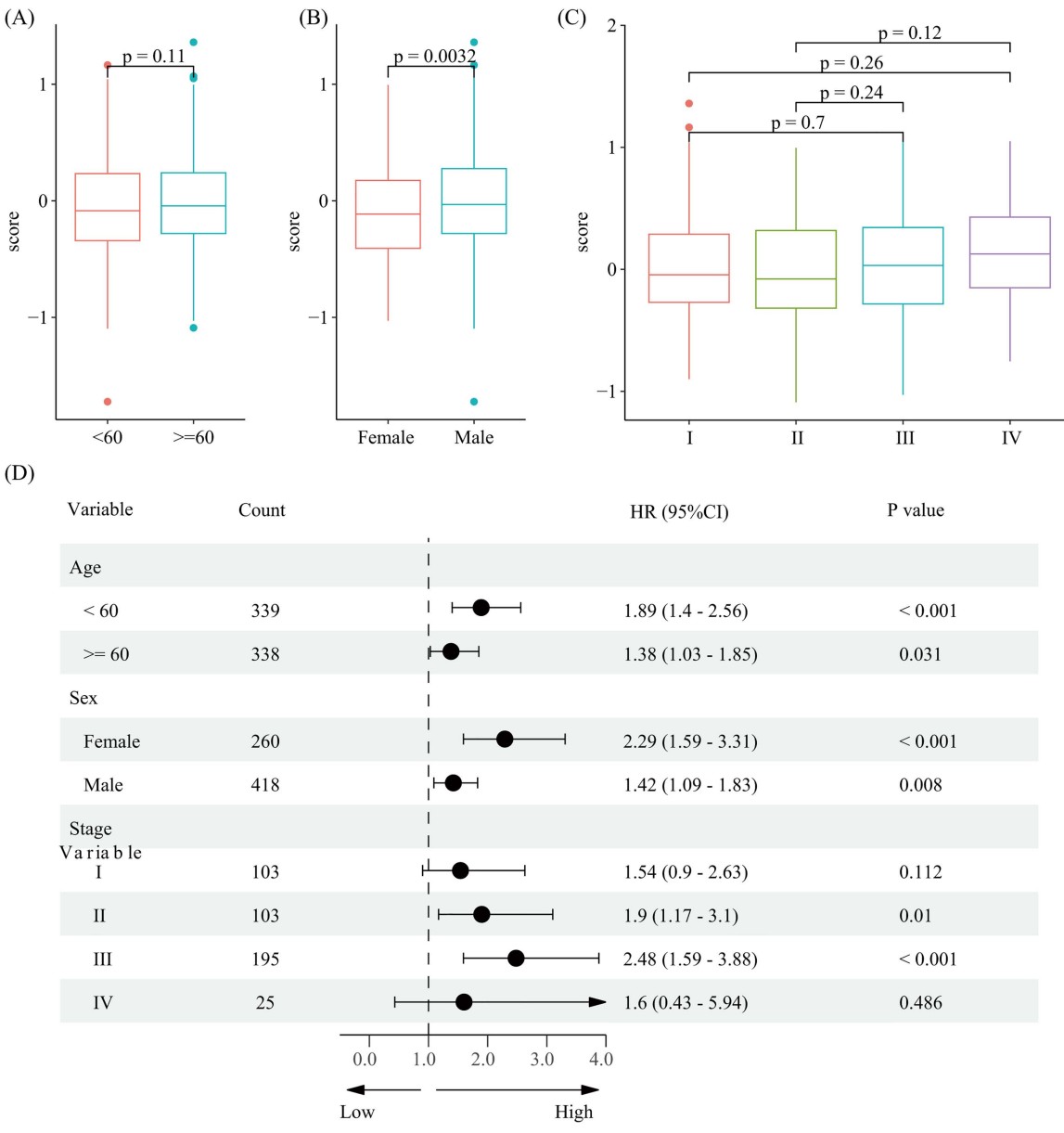

**Fig 3. Stratified analysis of the relationship between risk score and survival in metastatic melanoma patients.** (A) Risk scores stratified by age; (B) Risk scores stratified by gender; (C) Risk scores stratified by stage; (D) Prognostic analysis of high-risk and low-risk groups stratified by age, gender, and stage.

## Differences in immunogenicity indicators, immune cell abundance, and immunotherapy response among risk categories

We related various immunogenicity indicators among the high and low-risk categories. The consequences exhibited that, associated to the high-risk category, the low-risk category exhibited significantly lower levels of TMB (Tumor Mutational Burden), Tumor Purity, NTAI (Number of Telomeric Allelic Imbalances), LOH (Loss of Heterozygosity), LST (Large-scale State Transition), and Homologous Recombination Deficiency (HRD), while there was no statistically noteworthy variance in Somatic Copy Number Alteration (SCNA) among the two categories (Fig 6A).

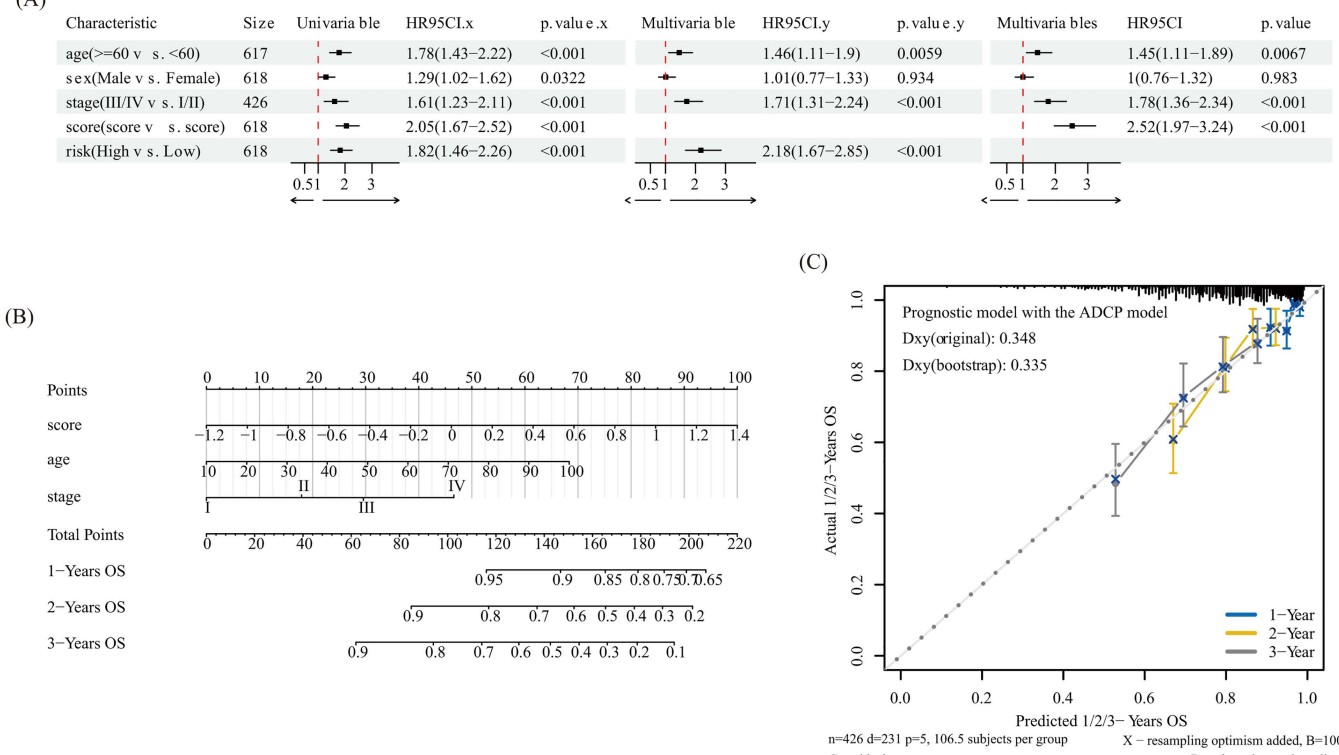

**Fig 4. Independent prognostic analysis of risk score and construction of the nomogram for metastatic melanoma.** (A) Univariate and multivariate Cox regression analysis of risk score and clinical characteristics; (B) Nomogram for predicting 1-, 2-, and 3-year OS based on risk score, age, and stage in metastatic melanoma patients; (C) Calibration curves for the nomogram predicting 1-, 2-, and 3-year OS.

To find out how the risk score and immune cell abundance in the tumour microenvironment of metastatic melanoma relate to one another, we performed a correlation study. The risk score was positively correlated with B cells naïve, NK cells resting, Macrophages M0, Macrophages M2, Myeloid dendritic cells activated, Mast cells activated, as well as Eosinophils, while the risk score was negatively correlated with B cells memory, T cells CD8 +, T cells CD4 + memory activated, T cells gamma delta, Macrophages M1, as well as Myeloid dendritic cells resting (Fig 6B). Out of the 22 immune cell types, two—macrophages M0 and T lymphocytes CD8+—showed notable variations in abundance between the high-risk and low-risk categories (Fig 6C).

Also evaluated whether risk score could predict immunotherapy response in metastatic melanoma patients. Patients were classified as Responders (R) and Non-Responders (NR) according to the study by He et al [28]. We discovered that the low-risk group had a much smaller number of NR than the high-risk category, and that the low-risk category's progression-free survival (PFS) was considerably higher than the high-risk category's (Fig 6D).

## Analysis of drug sensitivity

The top 12 medications in the high-risk and low-risk categories with notable variations in medication sensitivity are displayed in Fig 7. The findings suggested that these medications had greater sensitivity in the low-risk category since the IC50 values in that category were significantly reduced than those in the high-risk category.

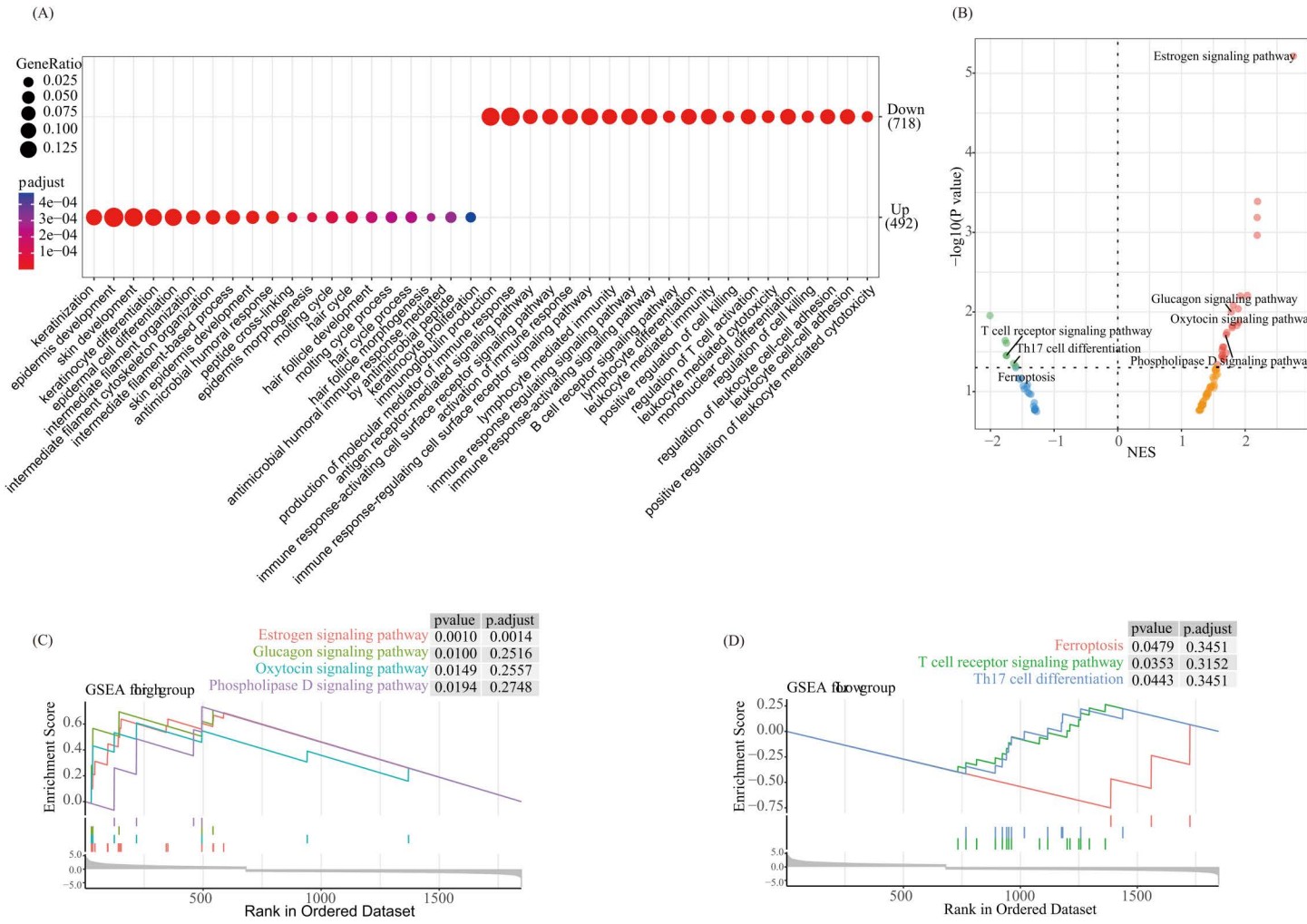

**Fig 5. Differences in biological functions and pathways between risk groups.** (A) GO annotation differences for differentially expressed genes between high-risk and low-risk groups; (B) GSEA analysis of differentially expressed genes between high-risk and low-risk groups; (C) GSEA enrichment curves for the high-risk group; (D) GSEA enrichment curves for the low-risk group. GO, Gene Ontology; GSEA, Gene Set Enrichment Analysis.

## Validation of risk genes

Although we have established a prognostic model for metastatic melanoma based on six ARGs (NDRG1, HRAS, KPNA2, ICAM1, DOCK10, and CDC20), to determine which gene plays a key role in melanoma metastasis, we further integrated single-cell transcriptomic data from primary and metastatic melanomas. Through filtering, dimensionality reduction clustering, and cell annotation, we identified eight cell subpopulations (Fig 8A). Notably, melanoma cells from metastatic tissues showed significantly higher DOCK10 expression compared to primary melanoma tissues (Fig 8B, C). Furthermore, we divided melanoma cells into high and low DOCK10 expression groups (Fig 8E). Through differential gene enrichment analysis using gene ontology (GO) (Fig 8D), we found that the high DOCK10 expression group was highly enriched in cadherin binding and focal adhesion pathways, which are related to cell adhesion.

To validate DOCK10's functional contribution to melanoma progression, we performed siRNA-mediated knockdown in A375 melanoma cells. qRT-PCR results confirmed that DOCK10 expression was significantly downregulated in A375 cells

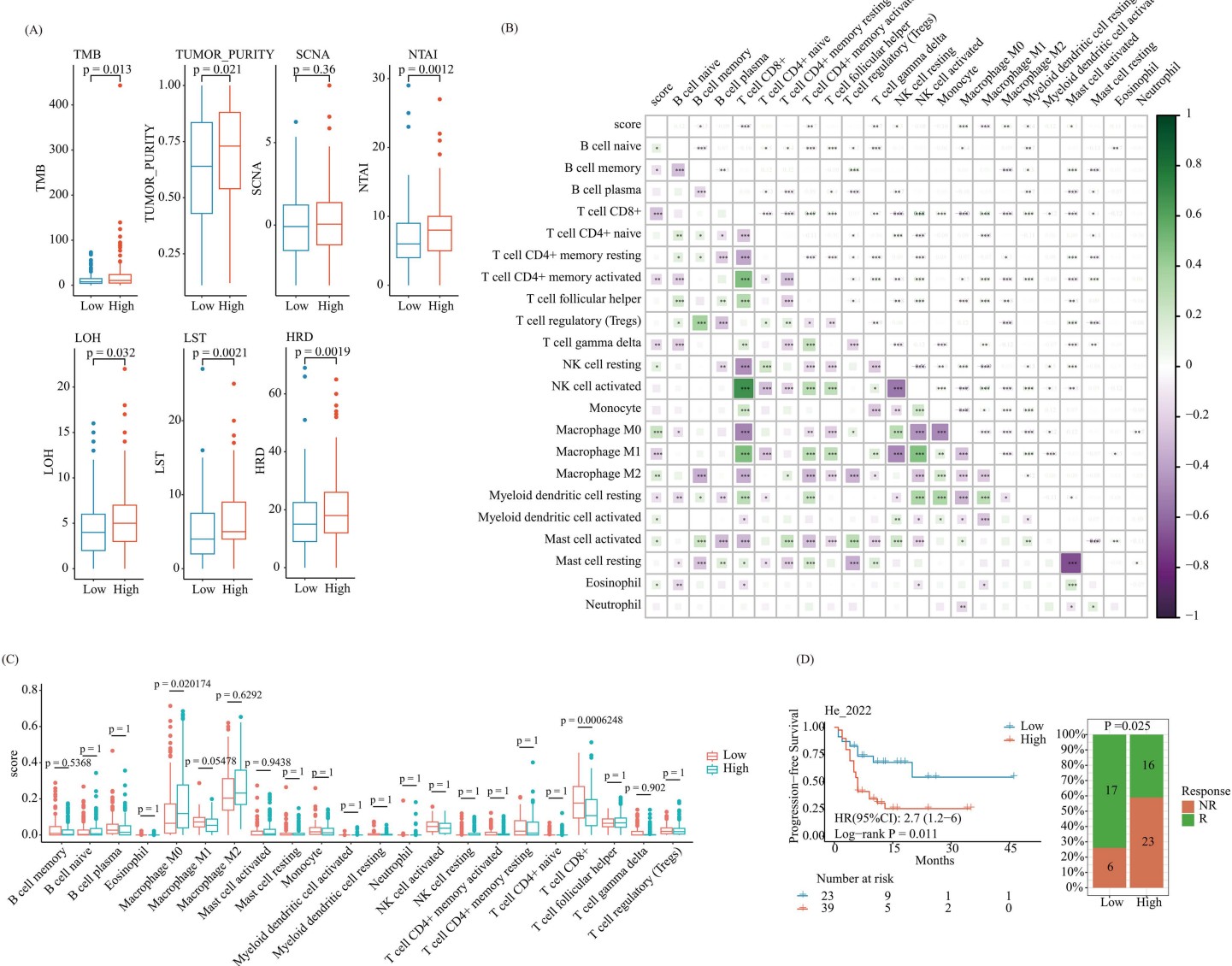

**Fig 6. Analysis of differences in immunogenicity indicators, immune cell abundance, and immunotherapy response between high-risk and low-risk groups.** (A) Analysis of immunogenicity indicators between high-risk and low-risk groups; (B) Correlation between risk score and immune cell infiltration; (C) Differences in immune cell abundance between high-risk and low-risk groups; (D) Differences in immunotherapy response between high-risk and low-risk groups.

transfected with DOCK10 siRNA compared to control siRNA (Fig 8F). The endothelial adhesion ability of A375 cells was assessed by fluorescence microscope. The results showed that silencing DOCK10 significantly inhibited the endothelial adhesion ability of A375 cells (Fig 8G). Ki67 was a marker of cell proliferation. Ki67 serves as a well-established marker of cell proliferation. Compared to control siRNA, transfection with DOCK10 siRNA markedly decreased Ki67 expression in A375 cells, suggesting a potential suppression of cell proliferation (Fig 8H). Invasion and migration are crucial processes in tumor cell motility. The invasive potential of A375 cells was evaluated using the Transwell assay. As depicted in Fig 8I, the knockdown of DOCK10 significantly inhibited the invasiveness of A375 cells. To further assess cell migration, wound

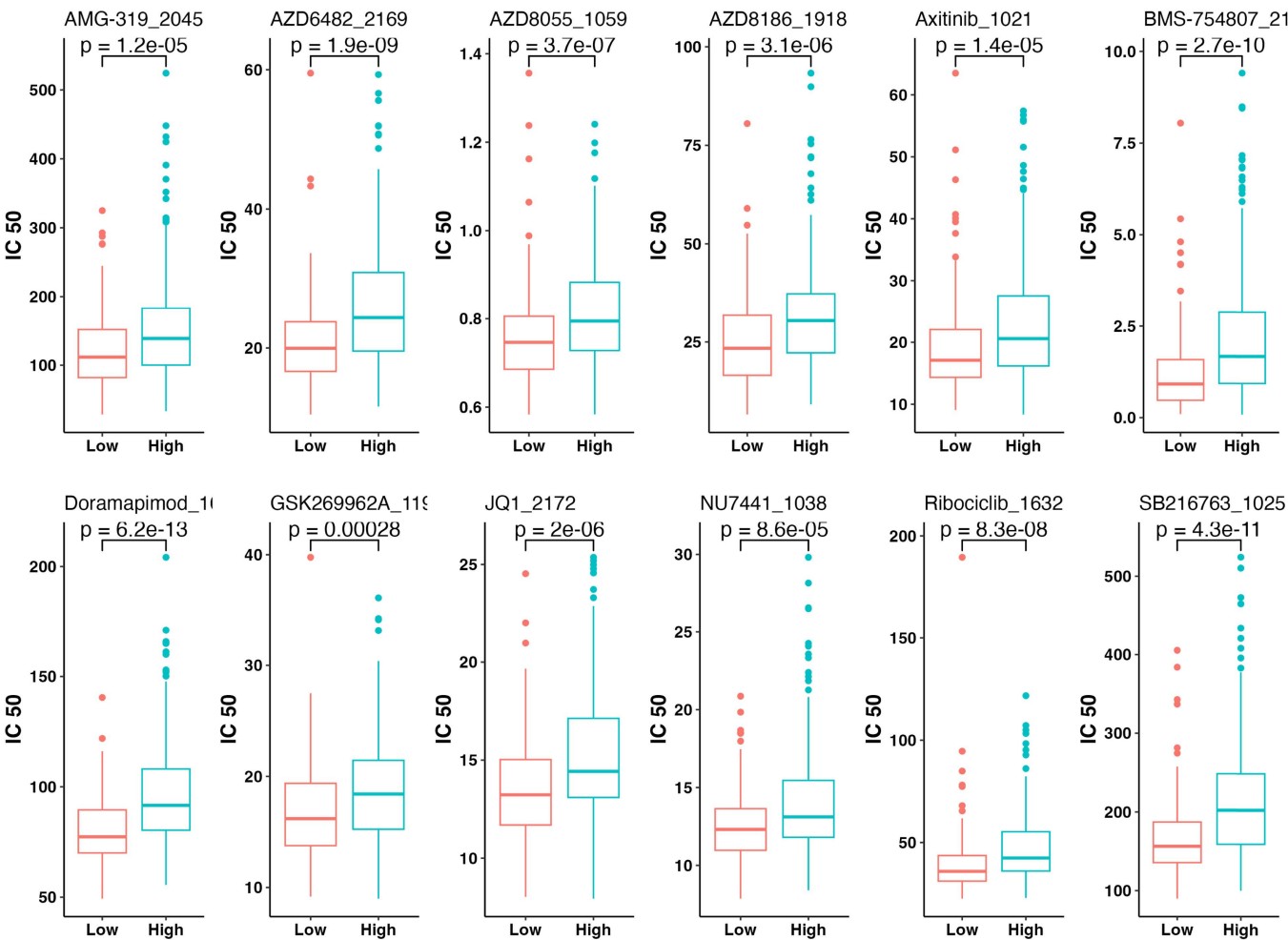

**Fig 7. Drug sensitivity differences between high-risk and low-risk groups.** Lower IC50 values indicate higher drug sensitivity in patients. IC50, Half-Maximal Inhibitory Concentration.

healing assays were conducted. Compared to control siRNA, silencing DOCK10 with siRNA notably impaired the migration of A375 cells (Fig 8J). The mRNA levels of CDH5, a protein implicated in cell migration and invasion, were reduced in A375 cells upon DOCK10 knockdown, as evidenced by qRT-PCR (Fig 8K). Western blot analysis was subsequently performed to measure the expression of proteins associated with cell proliferation, migration, and invasion. The findings revealed that DOCK10 siRNA markedly decreased the protein expression of Ki67, ITGB1, and CDH5 in A375 cells relative to control siRNA (Fig 8L). These comprehensive functional studies demonstrate that DOCK10 knockdown significantly impairs multiple hallmarks of melanoma progression, including proliferation, migration, and invasion. These evidences not only validate DOCK10 as a driver of metastatic behavior but also reinforce its candidacy as a therapeutic target within our ARG-based prognostic model.

## Discussion

Cutaneous melanoma, known for its high malignancy and metastatic potential, is considered the deadliest skin tumor [31]. Over the past few decades, melanoma prevalence has been gradually increasing worldwide, and cutaneous melanoma

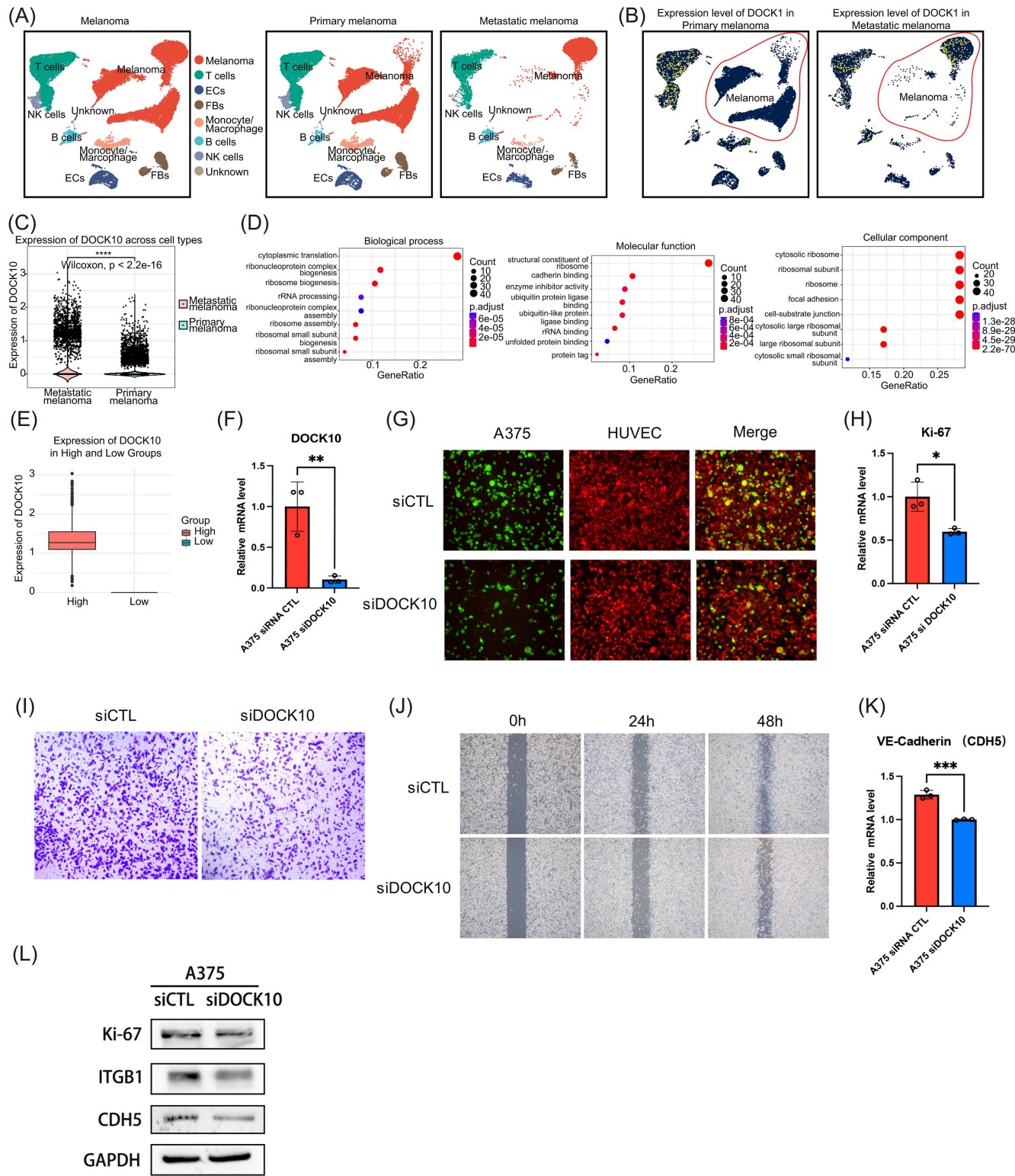

**Fig 8. Validation of Risk Genes.** (A) Annotation of cell clusters in metastatic and primary melanoma; (B) Expression levels of DOCK10 in different cell clusters in metastatic and primary melanoma; (C) Expression levels of DOCK10 in melanoma cells from metastatic and primary melanoma tissues; (D) Gene Ontology (GO) enrichment analysis; (E) Comparison of DOCK10 expression levels between high-risk and low-risk groups in metastatic melanoma

patients from the GSE189889 dataset; (F) mRNA levels of DOCK10 in A375 cells after siRNA transfection; (G) Immunofluorescence analysis of endothelial adhesion ability in A375 cells following DOCK10 silencing; (H) qRT-PCR analysis of Ki67 mRNA levels in A375 cells after DOCK10 silencing; (I) Transwell assay to assess the invasive ability of A375 cells following DOCK10 silencing; (J) Wound healing assay to evaluate the migration ability of A375 cells following DOCK10 silencing; (K) qRT-PCR analysis of CDH5 mRNA levels in A375 cells; (L) Western blot analysis of Ki67, ITGB1, and CDH5 protein expression levels in A375 cells. *$p < 0.05$, **$p < 0.01$, ***$p < 0.001$.

causes around 555,000 fatalities every year and around 232,100 new cases each year [32,33]. Therefore, developing effective prognostic models is essential for improving the clinical organization of metastatic melanoma and identifying potential therapeutic targets. Novel immune checkpoint inhibitors (ICIs), like anti-PD-L1 and anti-CTLA-4, have shown notable clinical success in treating melanoma in recent years [25,34]. ADCP serves as a terminal step in ICIs, wherein macrophages eliminate tumor cells following prior activation and recognition [19,35]. Leveraging macrophage-mediated ADCP as a promising and effective immune therapeutic strategy has shown anticancer activity within ICIs [36,37]. In this work, we developed a new ARG-based predictive model for metastatic melanoma. Results suggest that this model not only shows excellent predictive performance in prognosis but also has potential as a biomarker for immune response and drug sensitivity, promising broader utility in future research and clinical practice.

We identified six ARGs with prognostic value to construct the model: NDRG1, HRAS, KPNA2, ICAM1, DOCK10, and CDC20. The N-myc downregulated gene-1 is known to be expressed in numerous tumor types and exhibits dual roles in tumor biology [38]. In melanoma, NDRG1 is a powerful immune-related gene linked to the ferroptosis pathway [7,39,40]. HRAS and CDC20 are renowned participants in cell proliferation and mitotic checkpoint regulation. HRAS, a member of the RAS family, signals downstream of cell surface receptors to regulate cell proliferation, differentiation, and immune evasion [41–43]. Poor overall survival in patients with melanoma has been linked to high HRAS mRNA expression [43]. A ring-finger E3 ubiquitin ligase called cell division cycle 20 (CDC20) is essential for regulating the spindle assembly checkpoint (SAC) [44]. According to earlier research, increased CDC20 expression contributes to the development of tumours by enabling DNA-damaged cells to evade mitosis and apoptosis [45–47]. In melanoma, CDC20 is considered a core gene associated with prognosis [48]. KPNA2, ICAM1, and DOCK10 were identified for their roles in immune regulation and cell adhesion. Karyopherin α2 (KPNA2), one of 7 members of the nuclear protein α family, is known for its role in nuclear transport and is accompanying with tumor metastasis [49]. KPNA2 reportedly promotes proliferation, migration, and invasion in melanoma cells [50,51]. ICAM1 (Intercellular adhesion molecule 1) is essential for leukocyte adhesion and migration, facilitating immune cell infiltration [52]. In BAP1-mutant uveal melanoma, ITGB2-ICAM1 signaling promotes liver metastasis [53].

Using both internal and external datasets, we assessed the risk score prognostic model, which includes the six ADCP-related genes listed above. The findings showed that individuals with metastatic melanoma who had greater risk factors also have worse prognoses. An independent prognostic factor for metastatic melanoma may be the risk score, according to univariate and multivariate Cox regression models. In order to forecast the overall survival (OS) for individuals with metastatic melanoma at one, two, and three years, we then created a nomogram that included the risk score, age, and stage. The nomogram produced good OS prediction values for individuals with advanced melanoma, according to calibration plots.

Using GO and GSEA for enrichment analysis allows for an in-depth understanding of distinct biological processes and pathways among the high-risk and low-risk categories. The high-risk category showed enrichment in pathways related to Estrogen signaling, Glucagon signaling, Oxytocin signaling, and Phospholipase D signaling pathways, which are typically associated with tumor progression and metabolic adaptation [54–56]. In melanoma, the Estrogen signaling pathway has been shown to encourage immune suppression and reduce efficacy of immune checkpoint blockade therapy, while the Phospholipase D signaling pathway is related to cancer metastasis [57,58]. In contrast, the low-risk category was enriched

in immune-related pathways, counting Ferroptosis, T cell receptor signaling, and Th17 cell differentiation. Ferroptosis is a form of programmed cell death driven by iron-dependent lipid peroxidation [59]. As ferroptosis is closely related to regulating tumor growth and efficacy of radiotherapy, chemotherapy, and immunotherapy, it has recently gained attention as a potential therapeutic target for melanoma [60–62]. The T cell receptor signaling pathway and Th17 cell differentiation are crucial for maintaining a pro-inflammatory environment, which is essential for effective anti-tumor immunity [63–65]. These outcomes recommend that the enriched pathways in the low-risk category may enhance immune recognition and tumor control, supporting favorable prognostic characteristics.

Our results reveal significant differences in immunogenicity indicators among high-risk and low-risk categories. Sufferers in the low-risk category exhibited lower TMB, tumor purity, NTAI, LOH, LST, and HRD. Tumor purity refers to the proportion of tumor cells within the tumor microenvironment (TME). The lower tumor purity in the low-risk category suggests higher immune cell presence, potentially aiding in improved immune recognition and control of the tumor [66]. Increased levels of NTAI, LOH, LST, and HRD indicate greater genomic instability, which correlates with poorer prognosis. Although high TMB is traditionally associated with a better response to immunotherapy [67], our outcomes designate that combination of low-risk ADCP gene expression and moderate TMB might provide a more favorable immune environment, thereby reducing immune escape mechanisms.

We looked into the relationship between immune cell infiltration levels and risk scores. T cell, B cell, and activated NK cell counts—all of which are essential for anti-tumor immunity—were inversely connected with the risk score. Specifically, the presence of CD8 + T cells and memory T cells is typically accompanying with cytotoxic immune responses, suggesting that tumors with low-risk scores may elicit stronger immune responses [68–70]. Conversely, high-risk scores were connected with higher range of macrophages (M0 and M2 types) and activated mast cells, often linked to an immunosuppressive environment. In particular, M2 macrophages are well-known for promoting tumor growth and suppressing anti-tumor immunity, which may explain the poorer survival outcomes in the high-risk category [35,71]. This distinct immune profile underscores the value of the ARG-based prognostic model in reflecting the immune landscape of metastatic melanoma.

One notable aspect of this study is the drug sensitivity analysis, which indicates that low-risk individual may exhibit greater sensitivity to chemotherapeutic agents such as AMG-319 (PI3K δ inhibitor), AZD6482 (PI3K β inhibitor), AZD8055 (mTOR inhibitor), Axitinib, Doramapimod, and Ribociclib. Axitinib, a TKI (tyrosine kinase inhibitor) targeting VEGFR (vascular endothelial growth factor receptor), has been described to constrain cell proliferation in melanoma and enhance the proportion of CD8 + T cells, contributing to its anticancer effects [72]. Additionally, CD8 + T cells are meaningfully infiltrated in the low-risk category. Therefore, using Axitinib in the low-risk category could potentially yield unanticipated beneficial possessions by additional activating CD8 + T cells. Ribociclib, a CDK4/6 inhibitor, has been shown to regulate T cell immunity effectively, in addition to its impact on cell cycle progression [73,74]. A study demonstrated that Ribociclib promotes memory T cell formation and augments the effectiveness of PD-1 blockade treatment in the model of melanoma mouse [75].

## Advantages and limitations

This study has several strengths. Validation using multiple independent cohorts enhanced the robustness and generalizability of the ARG-based prognostic model. The integration of computational analysis with experimental validation, particularly the functional characterization of DOCK10, provides mechanistic insights that strengthen the biological foundation of our prognostic model. Additionally, the analyses of immunogenicity indicators, immune cell infiltration, and pathway enrichment provided a detailed understanding of ARGs and their clinical significance, which aids in prognostic assessment and the expansion of personalized therapeutic approaches for metastatic melanoma individual. However, some limitations remain. First, the cohorts employed in this investigations were obtained from public datasets, thus inevitably introduces tumor heterogeneity within and across patients. Potential comorbidities or prior treatments could affect immune status and treatment responses. The analysis relied heavily on a restricted number of samples in public datasets, highlighting the

need for larger sample sizes. While we provided functional validation for DOCK10, comprehensive experimental validation of all six genes in the model would further strengthen our findings. Future studies should focus on validating the remaining genes and elucidating their collective contribution to melanoma progression.

Although we have established a prognostic model for metastatic melanoma based on six ARGs (NDRG1, HRAS, KPNA2, ICAM1, DOCK10, and CDC20), in order to identify which gene plays a key role in melanoma metastasis, we further integrated single-cell transcriptomic data from primary melanoma and metastatic melanoma.Among these six genes, we found that DOCK10 is highly expressed in metastatic melanoma compared to primary melanoma. Therefore, DOCK10 may be an important driving factor in melanoma metastasis. Furthermore, we divided the melanoma cells into two groups based on the expression levels of DOCK10: high and low expression groups. Through differential gene enrichment analysis, we found that melanoma cells with high DOCK10 expression were highly enriched in the cadherin binding and focal adhesion pathways.Therefore, we hypothesize that DOCK10 plays an important role in the adhesion to endothelial cells during melanoma metastasis. We further knocked down DOCK10 in A375 cells using siRNA, and the results showed a significant decrease in the adhesion of A375 cells to endothelial cells. Additionally, Western blot experiments confirmed that the expression of adhesion-related proteins, ITGB1 and CDH5, was significantly reduced after DOCK10 knockdown in A375 cells. Moreover, the knockdown of DOCK10 resulted in a marked decrease in the proliferation, migration, and invasion abilities of the cells.Therefore, our experimental validation definitively demonstrated that DOCK10 knockdown impairs melanoma cell proliferation, migration, and invasion, confirming its functional significance in melanoma progression. This direct evidence of DOCK10's role in melanoma metastasis not only validates our computational predictions but also strengthens the biological foundation of our ARG-based prognostic model. Our scRNA-seq analysis provided additional insights into DOCK10's role by revealing its elevated expression in metastatic melanoma cells compared to primary melanoma cells at single-cell resolution. This cell-type specific expression pattern, combined with our functional validation showing DOCK10's critical role in melanoma cell behavior, establishes a mechanistic link between DOCK10 expression and melanoma progression. The convergence of computational prediction, single-cell expression analysis, and functional validation exemplifies the power of integrating multi-level evidence to understand prognostic gene signatures.

## Conclusion

In summary, we industrialized an ARG-grounded prognostic model that provides a promising tool for risk stratification in metastatic melanoma, with prognostic and therapeutic implications. By integrating ARGs with functional enrichment, immune characteristics, and drug sensitivity, this model offers a comprehensive perspective on tumor biology and actionable insights for personalized therapeutic strategies. The experimental validation of DOCK10 not only confirms its role as a key driver of melanoma metastasis but also validates our computational approach for identifying clinically relevant ADCP-related genes. This integration of bioinformatics prediction with functional validation establishes a paradigm for developing biologically grounded prognostic models that can guide both patient stratification and therapeutic target discovery in metastatic melanoma.

### Original images for blots and gels

Original, uncropped images of all blots and gels are provided in Supporting Information S4 Fig.

### Supporting information

**S1 Fig. Workflow diagram of the study.** The flowchart illustrates the systematic approach used in this research, including: (1) Data acquisition from TCGA and GEO databases; (2) Identification of differentially expressed genes between primary and metastatic melanoma; (3) Intersection with ADCP-related genes; (4) Feature selection using LASSO regression; (5) Construction of the six-gene prognostic model; (6) Validation in multiple cohorts; (7) Functional analyses including pathway enrichment, immune cell infiltration, and drug sensitivity prediction; (8) Experimental validation of DOCK10

through siRNA knockdown and functional assays. TCGA, The Cancer Genome Atlas; GEO, Gene Expression Omnibus; DEGs, Differentially Expressed Genes; ADCP, Antibody-Dependent Cellular Phagocytosis; LASSO, Least Absolute Shrinkage and Selection Operator; GO, Gene Ontology; GSEA, Gene Set Enrichment Analysis.
(PDF)

**S2 Fig. The heatmap of the 887 common DEGs.**
(PDF)

**S3 Fig. Expression levels of NDRG1,ICAM1,KPNA2,HRAS and CDC20 in different cell clusters in metastatic and primary melanoma.**
(PDF)

**S4 Fig. Original, uncropped images of all blots and gels.**
(PDF)

**S1 Table. Expression level of 887 DEGs in GSE46517.**
(XLSX)

## Author contributions

**Investigation:** Jiapeng He.

**Methodology:** Xiaolong Xu.

**Project administration:** Jianglin Zhang.

**Resources:** Xiaolong Xu.

**Supervision:** Jianglin Zhang.

**Validation:** Haiyan SUN.

**Visualization:** Haiyan SUN.

**Writing – original draft:** Junhao Chen.

**Writing – review & editing:** Jianglin Zhang.

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
