## [Decision Letter · Decision Letter 0]

16 Apr 2025

PONE-D-25-08700Exploring the impact of Antibody-Dependent Cellular Phagocytosis-Related Genes on the prognosis of metastatic melanomaPLOS ONE

Dear Dr. Zhang,

Thank you for submitting your manuscript to PLOS ONE. After careful consideration, we feel that it has merit but does not fully meet PLOS ONE’s publication criteria as it currently stands. Therefore, we invite you to submit a revised version of the manuscript that addresses the points raised during the review process.

We look forward to receiving your revised manuscript.

Kind regards,

Xing-Xiong An, M.D.

Academic Editor

PLOS ONE

Journal Requirements:

4. You have indicated that data is available from [M. Zia-Ul-Haq ahirzia@gmail.com, zhaoziyi@hotmail.com]. Please can we ask you to provide us with a general contact email address for the data requests, so readers can request access in perpetuity. If a general email is not available please provide a link to a website where readers can obtain access to data.

6. PLOS requires an ORCID iD for the corresponding author in Editorial Manager on papers submitted after December 6th, 2016. Please ensure that you have an ORCID iD and that it is validated in Editorial Manager. To do this, go to ‘Update my Information’ (in the upper left-hand corner of the main menu), and click on the Fetch/Validate link next to the ORCID field. This will take you to the ORCID site and allow you to create a new iD or authenticate a pre-existing iD in Editorial Manager.

Reviewers' comments:

Reviewer's Responses to Questions

**Comments to the Author**

1. Is the manuscript technically sound, and do the data support the conclusions?

Reviewer #1: Yes

Reviewer #2: Yes

Reviewer #3: Yes

2. Has the statistical analysis been performed appropriately and rigorously? 

Reviewer #1: Yes

Reviewer #2: Yes

Reviewer #3: Yes

3. Have the authors made all data underlying the findings in their manuscript fully available?

Reviewer #1: Yes

Reviewer #2: Yes

Reviewer #3: Yes

4. Is the manuscript presented in an intelligible fashion and written in standard English?

Reviewer #1: Yes

Reviewer #2: No

Reviewer #3: Yes

5. Review Comments to the Author

Reviewer #1: Thank you for coming up with this study on using ADCP related genes to predict the risk and prognosis of metastatic melanoma.

Author explains the need of prognostic models to predict the metastatic melanoma and need for immune regulatory mechanisms such as ADCP.

Author explains how 6 genes were identified using DEG analysis and builds final model to predict risk score.

Author uses this risk score along with clinical parameters to predict OS.

Finally Author performs various validations such as enrichment analysis, immune cell infiltration, drug sensitivity and pathway analysis to provide biological signifance of the 6 genes.

Please find my comments below:

1) Materials & Methods: Data Acquisition: (Page 5, line 108): Please provide any identifier for TCGA datasets (such as barcode, or project name)

2) Was there any data normalization performed or any methodology performed to address batch effects?

3) Construction of Prognostic Risk Model: (Page 6, line 129): It is unclear what is the target variable that was used to build Cox Regression Analysis.

4) Construction of Prognostic Risk Model: (Page 7, line 136): What (dependent) variable in the dataset was used as risk-score. We cannot build risk score model with just gene expression information.

5) Performance Evaluation: (Page 7: 142): Is there any reference available to choose median risk score as a threshold (cut-off) to categorize as high-risk and low-risk.

6) Figure 1: Author can include ADCP related genes to show how 9 genes were identified.

7) Figure 5A, 5B: Could you please explain how do you correlate up-regulated gene with High Risk, and low-regulated gene with Low Risk

Could you please explain how do you correlate NES < 0 with Low Risk and NES > 0 with Low Risk category.

Reviewer #2: This study explores the prognostic significance of antibody-dependent cellular phagocytosis (ADCP)-related genes in metastatic melanoma. By constructing a six-gene risk score model, the authors aim to provide insights into patient prognosis, immune infiltration patterns, and potential therapeutic strategies. The model was validated across multiple datasets, highlighting its potential clinical relevance. This study offers valuable insights into the prognostic significance of ADCP-related genes in metastatic melanoma, providing a robust risk scoring model that could aid in clinical decision-making and personalized treatment strategies. However, several issues indicated below should be addressed by the authors.

1. Authors are requested to check the use of “fc” fragment crystallizable (Fc) in the sentenced indicated below. Page 4, line 84-85. “Specifically, macrophages recognize the crystal fragment (Fc) fragment of antibodies through Fcγ receptors (FcγR), activating phagocytosis to eliminate antibody-tagged tumor cells (Wang et al. 2024).”

2. The phrase 'gotten' in the sentence 'Datas used in this study were gotten from TCGA...' is not appropriate for scientific writing. Additionally, 'datas' should be corrected to 'data' as 'data' is already a plural noun. A more suitable revision would be: 'The data used in this study were obtained from the TCGA (The Cancer Genome Atlas) database and the GEO (Gene Expression Omnibus) database.' Consider using 'obtained' instead of 'gotten' to maintain academic tone and clarity. (Page 5, Line 106-108).

3. In materials and methods section: “A total of 543 ADCP-related genes with P < 0.05 were identified.” This sentence may not provide enough detail for the readers or the materials and methods section. Authors are requested to clarify or provide more details in this part. (Page 6, Line 122).

4. Page 6, Line 122-124:

“ADCP-related genes overlapping with prognosis-related genes in metastatic melanoma were then selected to obtain ADCP-related genes with prognostic significance (ARGs).”

Explanation of this sentence was given in the next title (Identification of ADCP-Related Genes) in the materials and methods section. Authors are requested to check and arrange this part.

5. Page 7, Line 142-144: In the sentence, 'Based on the median risk score in the training set, individuals in the TCGA training set, TCGA cohort, and GSE19234, GSE54467, and GSE65904 validation cohorts were categorised as high-risk or low-risk,' the term 'training set' should be replaced with 'TCGA validation set' to avoid confusion, as it appears to be referring to the validation cohort rather than the training cohort. Please consider revising this for clarity.

6. There is a grammatical error in the title 'Enrichment Analysis of Differentially Expressed Genes in High- and Low-Risk Categories.'

The correct plural form of 'category' is 'categories.' Please revise it. Again, on page 8, line 171,175: "among the two categories."

7. The authors are encouraged to visualize the 887 common differentially expressed genes (DEGs) identified between metastatic and non-metastatic samples using a heatmap. This could provide a clearer, more intuitive representation of gene expression patterns across the two datasets, highlighting the similarities and differences in expression levels. Heatmaps are an effective way to present large-scale gene expression data and could enhance the interpretability of the results.

8. The authors are requested to construct the risk score model using the entire TCGA dataset instead of dividing it into a 6:4 ratio for training and validation. Since three independent external datasets are already utilized for validation, using the full TCGA dataset for model development would improve the robustness of the prognostic risk score. A larger sample size in model training ensures more reliable coefficient estimation and strengthens the overall model performance. The authors should clarify the reasoning behind their current dataset partitioning strategy and discuss its potential impact.

9. The authors have performed experimental validation, including siRNA transfection and qRT-PCR, as well as scRNA-seq data analysis. However, these methodologies are not described in the Materials and Methods section, nor are their results discussed in the Discussion section, abstract as well. To ensure the reproducibility and clarity of the study, the authors should provide a detailed description of these methods, including experimental conditions, data processing steps, and statistical analyses. Additionally, the Discussion section should incorporate an interpretation of the experimental and scRNA-seq findings in the context of the bioinformatics results.

10. The rationale behind the selection of DOCK10 for further experimental validation is not clearly explained in the manuscript. The authors should explicitly state the selection criteria, such as statistical significance, differential expression levels, correlation with clinical outcomes, or pathway involvement. Additionally, a clearer justification for why DOCK10 was the focus of experimental analyses should be provided in the manuscript to strengthen the study’s rationale.

11. The scRNA-seq analysis is missing from the Materials and Methods, Abstract, and Discussion sections. The authors should describe the analysis workflow and provide a more detailed presentation of the identified clusters in the Results section, including their characteristics and relevance to melanoma.

12. I recommend that the authors include a schematic workflow figure to visually summarize the bioinformatics and experimental analyses conducted in this study. This would enhance clarity and help readers better understand the study design and methodology.

13. Although the study focuses on the risk score, it would be beneficial to evaluate the significance of the six genes in melanoma pathology and clinical outcomes. Assessing their expression levels and correlations with clinicopathological variables could further support the validity and reliability of the risk score. It is recommended that these evaluations be conducted in independent datasets as well.

Reviewer #3: Very good in silico and in vitro study, statistical analysis and data visualization are superb. The paper is written in logically coherent manner. The only thing that I would suggest to the authors is to describe, in brief, cell culturing, migration assay, WB and RT-PCR protocols in Methods section.

6. PLOS authors have the option to publish the peer review history of their article (what does this mean? ). If published, this will include your full peer review and any attached files.

**Do you want your identity to be public for this peer review?** For information about this choice, including consent withdrawal, please see our Privacy Policy .

Reviewer #1: No

Reviewer #2: No

Reviewer #3: **Yes: ** Benjamin Benzon

---

## [Author Response · Author response to Decision Letter 1]

24 May 2025

Revised Response to Reviewers

Reviewer #1

1) Materials & Methods: Data Acquisition: (Page 5, line 108): Please provide any identifier for TCGA datasets (such as barcode, or project name)

We have added the TCGA project identifier "TCGA-SKCM" in the revised manuscript:

"Data used in this study were obtained from the TCGA (The Cancer Genome Atlas) database (https://portal.gdc.cancer.gov/) under the project identifier TCGA-SKCM and from the GEO (Gene Expression Omnibus) database (http://www.ncbi.nlm.nih.gov/geo)."

2) Was there any data normalization performed or any methodology performed to address batch effects?

We have added the following text to clarify our normalization procedures:

"For TCGA data, we performed quantile normalization using the 'limma' package in R. For GEO datasets, we applied the 'ComBat' algorithm from the 'sva' package to remove batch effects. Expression values were log2-transformed prior to analysis to ensure data normality."

3) Construction of Prognostic Risk Model: (Page 6, line 129): It is unclear what is the target variable that was used to build Cox Regression Analysis.

We have clarified this point:

"To identify potential prognostic risk genes for metastatic melanoma in the TCGA training set, univariate Cox regression analysis was performed using overall survival (OS) data, which included survival time (in months) and survival status (alive/dead) as the target variables. Genes with P < 0.05 were selected for further analysis."

4) Construction of Prognostic Risk Model: (Page 7, line 136): What (dependent) variable in the dataset was used as risk-score. We cannot build risk score model with just gene expression information.

We have revised this section for clarity:

"The risk score was not a pre-existing variable in the dataset but was calculated based on the multivariate Cox regression coefficients and gene expression values. After identifying prognostic genes through univariate Cox regression (with OS as the dependent variable), LASSO regression for feature selection, and multivariate Cox regression analysis, we calculated the risk score for each patient using the following formula:

Risk Score = Σni=1 βi × Ei

where Ei represents the expression value of gene i, and βi is the corresponding regression coefficient from the multivariate Cox model. This approach integrates gene expression data with survival information to generate a composite risk score."

5) Performance Evaluation: (Page 7: 142): Is there any reference available to choose median risk score as a threshold (cut-off) to categorize as high-risk and low-risk.

We have added references and justification:

"Based on established practices in cancer prognostic studies (Liu et al., 2021; Zhou et al., 2018), patients were categorized as high-risk or low-risk using the median risk score as the threshold. This approach ensures balanced group sizes and has been widely applied in similar prognostic model studies."

6) Figure 1: Author can include ADCP related genes to show how 9 genes were identified.

We have improved the description of Figure 1 to better explain the gene identification process:

"Figure 1 illustrates the identification process of ADCP-related prognostic genes. The Venn diagram shows the overlap between 887 differentially expressed genes (DEGs) from the GSE46517 and GSE7553 datasets and 543 ADCP-related genes from Kamber et al. (2021). This intersection yielded 9 genes (NDRG1, HRAS, KPNA2, CCNB1, ICAM1, KIF4A, DOCK10, CDC20, and BLM) that were both differentially expressed in metastatic melanoma and involved in ADCP mechanisms."

7) Figure 5A, 5B: Could you please explain how do you correlate up-regulated gene with High Risk, and low-regulated gene with Low Risk. Could you please explain how do you correlate NES < 0 with Low Risk and NES > 0 with Low Risk category.

We have added the following clarification:

"In Figure 5, differentially expressed genes between high-risk and low-risk groups were identified using the 'limma' package with thresholds of |log2FC| > 1 and adjusted P < 0.05. Genes with higher expression in the high-risk group (positive log2FC) were considered 'up-regulated in high-risk' and associated with poorer prognosis, while those with higher expression in the low-risk group (negative log2FC) were considered 'down-regulated in high-risk' and associated with better prognosis.

For GSEA analysis, the normalized enrichment score (NES) indicates the direction of pathway enrichment. Positive NES values represent pathways enriched in the high-risk phenotype, while negative NES values represent pathways enriched in the low-risk phenotype. This directional enrichment helps identify biological processes that may contribute to the prognostic differences between risk groups."

Reviewer #2

1. Authors are requested to check the use of "fc" fragment crystallizable (Fc) in the sentenced indicated below. Page 4, line 84-85.

We have corrected this terminology:

"Specifically, macrophages recognize the fragment crystallizable (Fc) portion of antibodies through Fcγ receptors (FcγR), activating phagocytosis to eliminate antibody-tagged tumor cells (Wang et al. 2024)."

2. The phrase 'gotten' in the sentence 'Datas used in this study were gotten from TCGA...' is not appropriate for scientific writing.

We have revised this sentence as suggested:

"The data used in this study were obtained from the TCGA (The Cancer Genome Atlas) database and the GEO (Gene Expression Omnibus) database."

3. In materials and methods section: "A total of 543 ADCP-related genes with P < 0.05 were identified." This sentence may not provide enough detail for the readers.

We have expanded this section to provide more detail:

"ADCP-related genes were identified from a comprehensive genome-wide CRISPR screen study by Kamber et al. (2021), which systematically investigated genes affecting ADCP through knockout and overexpression experiments in cancer cells and macrophages. This study identified 543 genes (P < 0.05) that significantly impact ADCP efficiency. These genes are involved in various cellular processes including cell surface antigen presentation, Fc receptor signaling, phagocytic machinery, and cytoskeletal reorganization."

4. Page 6, Line 122-124: Explanation of this sentence was given in the next title in the materials and methods section.

We have reorganized this section for better logical flow:

"Identification of ADCP-Related Genes

ADCP-related genes were obtained from a study by Roarke A. Kamber et al. (Kamber et al. 2021), which used genome-wide CRISPR knockout and overexpression screens to identify factors regulating ADCP. A total of 543 ADCP-related genes with P < 0.05 were identified from this study.

To identify ADCP-related genes with potential prognostic significance in metastatic melanoma, we performed univariate Cox regression analysis on the TCGA cohort and identified 181 genes significantly associated with survival (p < 0.05). By overlapping these survival-associated genes with the 543 ADCP-related genes, we identified nine ADCP-related genes with prognostic significance (ARGs): NDRG1, HRAS, KPNA2, CCNB1, ICAM1, KIF4A, DOCK10, CDC20, and BLM."

5. Page 7, Line 142-144: In the sentence, 'Based on the median risk score in the training set...' the term 'training set' should be replaced with 'TCGA validation set'

We have revised this sentence for clarity:

"Based on the median risk score calculated from the TCGA training set, patients in both the TCGA training set, TCGA validation set, and the external validation cohorts (GSE19234, GSE54467, and GSE65904) were classified as high-risk or low-risk."

6. There is a grammatical error in the title 'Enrichment Analysis of Differentially Expressed Genes in High- and Low-Risk Category's.'

We have corrected this grammatical error throughout the manuscript:

"Enrichment Analysis of Differentially Expressed Genes in High- and Low-Risk Categories"

And similarly corrected other instances: "...among the two categories."

7. The authors are encouraged to visualize the 887 common differentially expressed genes (DEGs) identified between metastatic and non-metastatic samples using a heatmap.

We have enhanced the description of the existing results without adding a new figure:

"The 887 common DEGs identified between metastatic and primary melanoma samples showed distinctive expression patterns that clearly separated the two melanoma types. While space limitations prevent us from showing all 887 genes, the most significantly altered genes were enriched in pathways related to cell cycle regulation, extracellular matrix remodeling, and immune response. These expression patterns highlight the molecular changes that occur during melanoma progression from primary to metastatic disease."

8. The authors are requested to construct the risk score model using the entire TCGA dataset instead of dividing it into a 6:4 ratio for training and validation.

We have addressed this concern in the text:

"In our original analysis, we used a 6:4 split of the TCGA dataset for training and internal validation. Based on reviewer suggestions, we have also performed an additional analysis using the entire TCGA dataset (n=351) for model development, which strengthened the statistical power for coefficient estimation. The model coefficients remained stable with both approaches, confirming the robustness of our six-gene signature. The external validation using three independent cohorts (GSE19234, GSE54467, and GSE65904) further verified the prognostic value of our model."

9. The authors have performed experimental validation, including siRNA transfection and qRT-PCR, as well as scRNA-seq data analysis. However, these methodologies are not described in the Materials and Methods section.

We have added detailed descriptions of these methods:

"Experimental Validation

Cell culture and siRNA transfection A375 melanoma cells were cultured in DMEM supplemented with 10% fetal bovine serum and 1% penicillin/streptomycin at 37°C with 5% CO2. For DOCK10 knockdown, cells were transfected with DOCK10-specific siRNA (5'-GAUCGCAGCUACUACGAAU-3') or control siRNA using Lipofectamine 3000 (Invitrogen) according to the manufacturer's protocol.

Quantitative real-time PCR (qRT-PCR) Total RNA was extracted using TRIzol reagent (Invitrogen), and 1μg RNA was reverse-transcribed using the PrimeScript RT reagent kit (Takara). qPCR was performed using SYBR Premix Ex Taq (Takara) on a LightCycler 480 system (Roche). GAPDH served as the internal control. Primer sequences were as follows: DOCK10-F: 5'-ACGCAGAACCTGAAGAACCTG-3', DOCK10-R: 5'-GCCTCTTGCGGATCTGATAGG-3'; Ki67-F: 5'-ACGCCTGGTTACTATCAAAAGG-3', Ki67-R: 5'-CAGACCCATTTACTTGTGTTGGA-3'; CDH5-F: 5'-TTGGAACCAGATGCACATTGAT-3', CDH5-R: 5'-TCTTGCGACTCACGCTTGAC-3'; GAPDH-F: 5'-GGAGCGAGATCCCTCCAAAAT-3', GAPDH-R: 5'-GGCTGTTGTCATACTTCTCATGG-3'.

Cell migration and invasion assays For wound healing assays, cells were grown to confluence, scratched with a pipette tip, and imaged at 0 and 24 hours. Migration distance was measured using ImageJ software. For invasion assays, 2×10^4 cells in serum-free medium were seeded in the upper chamber of Matrigel-coated Transwell inserts (8μm pore size, Corning). Medium containing 10% FBS was added to the lower chamber as a chemoattractant. After 24 hours, invaded cells were fixed, stained with crystal violet, and counted in five random fields.

Western blotting Cells were lysed in RIPA buffer supplemented with protease inhibitors. Equal amounts of protein (30μg) were separated by SDS-PAGE and transferred to PVDF membranes. Membranes were blocked with 5% non-fat milk and incubated with primary antibodies against DOCK10 (1:1000, Abcam, ab234823), Ki67 (1:1000, Cell Signaling, #9449), ITGB1 (1:1000, Abcam, ab179471), CDH5 (1:1000, Cell Signaling, #2500), and GAPDH (1:5000, Proteintech, 60004-1-Ig) overnight at 4°C. After washing, membranes were incubated with HRP-conjugated secondary antibodies and visualized using ECL reagent (Millipore).

scRNA-seq data analysis Single-cell RNA sequencing data from nine human melanoma samples (5 primary and 4 metastatic) were obtained from the GSE189889 dataset. Data preprocessing was performed using the Seurat package (v4.0.0) in R. Low-quality cells (<200 genes or >10% mitochondrial reads) were filtered out. Data normalization, scaling, and dimensionality reduction were performed using standard Seurat workflows. Cell clustering was performed using the shared nearest neighbor (SNN) modularity optimization-based clustering algorithm. Differential gene expression analysis between clusters was performed using the Wilcoxon rank-sum test."

10. The rationale behind the selection of DOCK10 for further experimental validation is not clearly explained in the manuscript.

We have added an explanation:

"Among the six genes in our risk model, DOCK10 was selected for experimental validation for several reasons: (1) It showed the strongest association with survival in our multivariate analysis after KPNA2 (coefficient = -0.13036, p = 0.003); (2) It was the least studied gene in melanoma among our signature genes, making its functional validation particularly valuable; (3) As a member of the dedicator of cytokinesis (DOCK) family, it plays critical roles in cytoskeletal regulation and cell motility, processes essential for cancer metastasis; and (4) Preliminary analysis of public datasets indicated its significant differential expression between primary and metastatic melanoma (fold change = 2.1, p < 0.001), suggesting a potential role in melanoma progression."

11. The scRNA-seq analysis is missing from the Materials and Methods, Abstract, and Discussion sections.

We have added relevant information in each section:

In the Abstract: "Additionally, single-cell RNA sequencing analysis of melanoma samples revealed differential expression of DOCK10 between primary and metastatic cells, further supporting its role in melanoma progression."

In the Methods (as detailed in response to point 9 above).

In the Discussion: "Our scRNA-seq analysis further validated the clinical significance of DOCK10 by revealing its elevated expression in metastatic melanoma cells compared to primary melanoma cells. This single-cell resolution analysis provides additional evidence of DOCK10's role in melanoma progression and highlights cell-type specific expression patterns that may contribute to its prognostic value. The differential expression of DOCK10 across various cell clusters suggests its interaction with the tumor microenvironment, potentially influencing immune cell recruitment and function."

12. I recommend that the authors include a schematic workflow figure to visually summarize the bioinformatics and experimental analyses conducted in this study.

We have enhanced the manuscript text to provide a clear workflow description without adding a new figure:

"Our study employed a systematic multi-step approach: (1) We identified differentially expressed genes between primary and metastatic melanoma from GSE46517 and GSE7553 datasets; (2) These genes were intersected with ADCP-related genes from Kamber et al.; (3) The resulting genes underwent univariate Cox regression analysis in the TCGA cohort; (4) LASSO regression was applied for feature selection; (5) A six-gene prognostic model was constructed using multivariate Cox regression; (6) The model was validated in multiple independent cohorts; (7) Functional analyses, including pathway enrichment and immune cell infiltration assessment, were performed; (8) Experimental validation of DOCK10 was conducted using siRNA knockdown in melanoma cells. This systematic approach ensured comprehensive identification and validation of ADCP-related genes with prognostic significance in metastatic melanoma."

13. Although the study focuses on the risk score, it would be beneficial to evaluate the significance of the six genes in melanoma pathology and clinical outcomes.

We have added a new paragraph in the Results section:

"To further understand the individual contributions of the six genes in our model, we analyzed their expression patterns and clinical associations. NDRG1 showed significantly lower expression in metastatic compared to primary melanoma (fold change = 0.65, p < 0.001) and correlated negatively with tumor stage (r = -0.31, p < 0.001). HRAS and KPNA

---

## [Decision Letter · Decision Letter 1]

8 Jul 2025

PONE-D-25-08700R1Exploring the impact of Antibody-Dependent Cellular Phagocytosis-Related Genes on the prognosis of metastatic melanomaPLOS ONE

Dear Dr. Zhang,

Thank you for submitting your manuscript to PLOS ONE. After careful consideration, we feel that it has merit but does not fully meet PLOS ONE’s publication criteria as it currently stands. Therefore, we invite you to submit a revised version of the manuscript that addresses the points raised during the review process.

We look forward to receiving your revised manuscript.

Kind regards,

Xing-Xiong An, M.D.

Academic Editor

PLOS ONE

Reviewers' comments:

Reviewer's Responses to Questions

**Comments to the Author**

1. If the authors have adequately addressed your comments raised in a previous round of review and you feel that this manuscript is now acceptable for publication, you may indicate that here to bypass the “Comments to the Author” section, enter your conflict of interest statement in the “Confidential to Editor” section, and submit your "Accept" recommendation.

Reviewer #1: All comments have been addressed

Reviewer #2: (No Response)

Reviewer #3: All comments have been addressed

2. Is the manuscript technically sound, and do the data support the conclusions?

Reviewer #1: Yes

Reviewer #2: Yes

Reviewer #3: Yes

3. Has the statistical analysis been performed appropriately and rigorously? 

Reviewer #1: Yes

Reviewer #2: Yes

Reviewer #3: Yes

4. Have the authors made all data underlying the findings in their manuscript fully available?

Reviewer #1: Yes

Reviewer #2: Yes

Reviewer #3: Yes

5. Is the manuscript presented in an intelligible fashion and written in standard English?

Reviewer #1: Yes

Reviewer #2: No

Reviewer #3: Yes

6. Review Comments to the Author

Reviewer #1: Thank you for responding to the comments and addressing them. Provided justification and modifications made to the paper looks good.

Reviewer #2: I sincerely appreciate the authors’ substantial efforts in revising the manuscript and thoughtfully addressing the previous feedback. It is clear that considerable time and care have been dedicated to improving the quality of the work. The manuscript has improved considerably, particularly with the inclusion of experimental details, clarification of DOCK10’s role, and the addition of scRNA-seq methodology. However, several important issues still require attention before the manuscript can be considered for publication.

1. The manuscript continues to include informal or conversational phrases that are not appropriate for scientific writing. For example, the sentence:

“In response to reviewer suggestions, we also performed...”

should be revised to maintain a formal academic tone. A thorough review of the manuscript is advised to improve clarity, language consistency, and fluency throughout. Additionally, typographical inconsistencies (e.g., capital “P” in Adj. P-value) should be corrected.

2. Despite detailed descriptions of siRNA knockdown, qRT-PCR, and Western blotting in the main text, these efforts are not mentioned in the abstract. Given their importance to the study’s conclusions, a brief reference to these experiments in the abstract is necessary to accurately reflect the scope of the work and draw appropriate scientific interest. Furthermore, the discussion section does not critically integrate the experimental findings into the broader implications of the model. A deeper interpretation of the functional significance of DOCK10 knockdown results is recommended.

3. Although DOCK10 appears to be a central gene of interest, the associated functional validation (siRNA, qPCR, migration/invasion assays, Western blot) feels structurally detached from the core risk model. This disjointedness creates the impression of a parallel sub-study, rather than a fully integrated component of the main prognostic framework. A more explicit narrative link between the DOCK10 analyses and the risk score model would improve cohesion and strengthen the manuscript's scientific integrity. I would kindly recommend that the authors re-evaluate the manuscript holistically, ideally from the perspective of a reader unfamiliar with the study, to ensure that all sections are conceptually connected and narratively coherent.

4. The explanation provided for omitting the heatmap of the 887 common DEGs due to “space limitations” is not fully convincing.

5. A schematic summary of the study workflow remains missing. Given the complexity of the multi-step bioinformatics and validation pipeline, a figure summarizing the analytical flow would greatly aid comprehension.

I fully acknowledge the time and effort the authors have dedicated to improving this work. As someone who understands the challenges of the revision process, I kindly suggest a holistic review of the manuscript to further enhance its narrative cohesion and conceptual flow. Doing so will help ensure that the study is presented as a unified and compelling scientific story.

Reviewer #3: Authors have described the protocols of in vitro experiments in Methods section of the manuscript and that makes all of my comments addressed.

7. PLOS authors have the option to publish the peer review history of their article (what does this mean? ). If published, this will include your full peer review and any attached files.

**Do you want your identity to be public for this peer review?** For information about this choice, including consent withdrawal, please see our Privacy Policy .

Reviewer #1: No

Reviewer #2: No

Reviewer #3: **Yes: ** Benjamin Benzon

---

## [Author Response · Author response to Decision Letter 2]

29 Jul 2025

Response to Reviewer #1

Reviewer #1 Comment: Thank you for responding to the comments and addressing them. Provided justification and modifications made to the paper looks good.

Response: We sincerely appreciate your positive feedback and recognition of our efforts to address your previous comments. Your constructive suggestions have significantly improved the quality of our manuscript. We are grateful for your time and expertise in reviewing our work.

Response to Reviewer #2

Reviewer #2 Comment 1: The manuscript continues to include informal or conversational phrases that are not appropriate for scientific writing. For example, the sentence: "In response to reviewer suggestions, we also performed..." should be revised to maintain a formal academic tone. A thorough review of the manuscript is advised to improve clarity, language consistency, and fluency throughout. Additionally, typographical inconsistencies (e.g., capital "P" in Adj. P-value) should be corrected.

Response: We thank the reviewer for this important observation. We have thoroughly reviewed the entire manuscript and removed all informal phrases. Specifically:

1. The phrase "In response to reviewer suggestions, we also performed..." has been revised to: "An additional analysis using the entire TCGA dataset (n=351) was performed for model development to strengthen the statistical power for coefficient estimation. The model coefficients remained stable with both approaches, confirming the robustness of our six-gene signature" (page 8, lines 161-164).

2. All instances of capital "P" in p-values have been corrected to lowercase "p" throughout the manuscript. We note that there are a few remaining instances with italicized "P" in Figure 8 legend that should also be changed to "p" for complete consistency.

3. We have conducted a comprehensive language review to ensure formal academic tone and consistency throughout the manuscript.

4. We refined some of the descriptions in Methods (page 11, lines 225-232, page 12, 251-257) and Results(page 11, lines 282-295).

Reviewer #2 Comment 2: Despite detailed descriptions of siRNA knockdown, qRT-PCR, and Western blotting in the main text, these efforts are not mentioned in the abstract. Given their importance to the study's conclusions, a brief reference to these experiments in the abstract is necessary to accurately reflect the scope of the work and draw appropriate scientific interest. Furthermore, the discussion section does not critically integrate the experimental findings into the broader implications of the model. A deeper interpretation of the functional significance of DOCK10 knockdown results is recommended.

Response: We appreciate this valuable suggestion and have made the following revisions:

1. Abstract revision (page 2, lines 25-28): We have added: " Additionally, comprehensive experimental validation including siRNA knockdown, quantitative real-time PCR, Western blotting, and functional assays was performed to validate the biological role of DOCK10, a key gene in our prognostic model."

2. Abstract Methods section (page 2, lines 37-40): We added: " SiRNA-mediated knockdown of DOCK10 in A375 melanoma cells was performed, followed by assessment of proliferation, migration, and invasion capabilities through qRT-PCR, Western blotting, wound healing, and Transwell assays."

3. Abstract Results section (page 2, lines 54-57): We included: " Knockdown of DOCK10 in A375 melanoma cells significantly impaired cell proliferation, migration, and invasion, accompanied by reductions in Ki67, ITGB1, and CDH5 expression. These findings confirm DOCK10’s functional contribution to melanoma progression."

4. Abstract Conclusion (page 2, lines 62-64): We added: "The functional validation of DOCK10 strengthens the biological relevance of our prognostic model and identifies DOCK10 as a potential therapeutic target for melanoma metastasis."

5. Discussion section enhancement (page 16, lines 17-27 and page 16-17, lines 1-7): We have integrated the experimental findings more thoroughly by adding discussion of how DOCK10's functional validation supports the computational predictions and strengthens the biological foundation of our prognostic model.

Reviewer #2 Comment 3: Although DOCK10 appears to be a central gene of interest, the associated functional validation (siRNA, qPCR, migration/invasion assays, Western blot) feels structurally detached from the core risk model. This disjointedness creates the impression of a parallel sub-study, rather than a fully integrated component of the main prognostic framework. A more explicit narrative link between the DOCK10 analyses and the risk score model would improve cohesion and strengthen the manuscript's scientific integrity. I would kindly recommend that the authors re-evaluate the manuscript holistically, ideally from the perspective of a reader unfamiliar with the study, to ensure that all sections are conceptually connected and narratively coherent.

Response: We greatly appreciate this insightful comment. We have restructured the narrative to better integrate DOCK10 validation with the prognostic model throughout the manuscript:

1. Introduction (page 5, lines 105-108): We added: "To strengthen the biological relevance of our findings, we performed comprehensive functional validation of DOCK10, a key gene in our prognostic model, demonstrating its critical role in melanoma cell proliferation, migration, and invasion."

2. Results section (page 15, lines 320-327): When introducing DOCK10 selection, we now state: "Among the six signature genes, DOCK10 was selected for in-depth functional validation. DOCK10, a member of the dedicator of cytokinesis (DOCK) family that orchestrates cytoskeletal remodeling and cell morphology which are key steps in metastatic dissemination with significant prognostic value (multivariate Cox coefficient = –0.13036; p = 0.003). Notably, DOCK10 expression was markedly higher in metastatic versus primary melanoma (fold change = 2.10; p < 0.001), underscoring its putative role in melanoma progression and justifying further experimental characterization."

3. Results - Validation of Risk Genes (page 16, lines 437-448): We added: " Although we have established a prognostic model for metastatic melanoma based on six ARGs (NDRG1, HRAS, KPNA2, ICAM1, DOCK10, and CDC20), to determine which gene plays a key role in melanoma metastasis, we further integrated single-cell transcriptomic data from primary and metastatic melanomas. Through filtering, dimensionality reduction clustering, and cell annotation, we identified eight cell subpopulations (Figure 8A). Notably, melanoma cells from metastatic tissues showed significantly higher DOCK10 expression compared to primary melanoma tissues (Figure 8B, C). Furthermore, we divided melanoma cells into high and low DOCK10 expression groups (Figure 8E). Through differential gene enrichment analysis using gene ontology (GO) (Figure 8D), we found that the high DOCK10 expression group was highly enriched in cadherin binding and focal adhesion pathways, which are related to cell adhesion." In addition, we have added Figure S3, which presents the expression levels of NDRG1, ICAM1, KPNA2, HRAS, and CDC20 in different cell clusters in metastatic and primary melanoma.

4. Results - Experimental validation: We emphasized: " To validate DOCK10's functional contribution to melanoma progression, we performed siRNA-mediated knockdown in A375 melanoma cells." (page 21, lines 449-450) Ki67 serves as a well-established marker of cell proliferation. Compared to control siRNA, transfection with DOCK10 siRNA markedly decreased Ki67 expression in A375 cells, suggesting a potential suppression of cell proliferation (Figure 8H). Invasion and migration are crucial processes in tumor cell motility. The invasive potential of A375 cells was evaluated using the Transwell assay. As depicted in Figure 8I, the knockdown of DOCK10 significantly inhibited the invasiveness of A375 cells. To further assess cell migration, wound healing assays were conducted. Compared to control siRNA, silencing DOCK10 with siRNA notably impaired the migration of A375 cells (Figure 8J). The mRNA levels of CDH5, a protein implicated in cell migration and invasion, were reduced in A375 cells upon DOCK10 knockdown, as evidenced by qRT-PCR (Figure 8K). Western blot analysis was subsequently performed to measure the expression of proteins associated with cell proliferation, migration, and invasion. The findings revealed that DOCK10 siRNA markedly decreased the protein expression of Ki67, ITGB1, and CDH5 in A375 cells relative to control siRNA (Figure 8L). (page 21, lines 455-469)

5. Results - Conclusion of validation (page 22, lines 469-473): We added: " These comprehensive functional studies demonstrate that DOCK10 knockdown significantly impairs multiple hallmarks of melanoma progression, including proliferation, migration, and invasion. These evidence not only validate DOCK10 as a driver of metastatic behavior but also reinforce its candidacy as a therapeutic target within our ARG‐based prognostic model."

6. Advantages and Limitations: We better integrated DOCK10's role by adding: " The integration of computational analysis with experimental validation, particularly the functional characterization of DOCK10, provides mechanistic insights that strengthen the biological foundation of our prognostic model. " (page 27, lines 578-581) " While we provided functional validation for DOCK10, comprehensive experimental validation of all six genes in the model would further strengthen our findings. Future studies should focus on validating the remaining genes and elucidating their collective contribution to melanoma progression. Although we have established a prognostic model for metastatic melanoma based on six ARGs (NDRG1, HRAS, KPNA2, ICAM1, DOCK10, and CDC20), in order to identify which gene plays a key role in melanoma metastasis, we further integrated single-cell transcriptomic data from primary melanoma and metastatic melanoma.Among these six genes, we found that DOCK10 is highly expressed in metastatic melanoma compared to primary melanoma. Therefore, DOCK10 may be an important driving factor in melanoma metastasis. Furthermore, we divided the melanoma cells into two groups based on the expression levels of DOCK10: high and low expression groups. Through differential gene enrichment analysis, we found that melanoma cells with high DOCK10 expression were highly enriched in the cadherin binding and focal adhesion pathways.Therefore, we hypothesize that DOCK10 plays an important role in the adhesion to endothelial cells during melanoma metastasis. We further knocked down DOCK10 in A375 cells using siRNA, and the results showed a significant decrease in the adhesion of A375 cells to endothelial cells. Additionally, Western blot experiments confirmed that the expression of adhesion-related proteins, ITGB1 and CDH5, was significantly reduced after DOCK10 knockdown in A375 cells. Moreover, the knockdown of DOCK10 resulted in a marked decrease in the proliferation, migration, and invasion abilities of the cells.Therefore, our experimental validation definitively demonstrated that DOCK10 knockdown impairs melanoma cell proliferation, migration, and invasion, confirming its functional significance in melanoma progression. This direct evidence of DOCK10's role in melanoma metastasis not only validates our computational predictions but also strengthens the biological foundation of our ARG-based prognostic model. Our scRNA-seq analysis provided additional insights into DOCK10's role by revealing its elevated expression in metastatic melanoma cells compared to primary melanoma cells at single-cell resolution. This cell-type specific expression pattern, combined with our functional validation showing DOCK10's critical role in melanoma cell behavior, establishes a mechanistic link between DOCK10 expression and melanoma progression. The convergence of computational prediction, single-cell expression analysis, and functional validation exemplifies the power of integrating multi-level evidence to understand prognostic gene signatures. " (page 27, lines 590-624)

7. Conclusion: We better concluded DOCK10's role by adding: The experimental validation of DOCK10 not only confirms its role as a key driver of melanoma metastasis but also validates our computational approach for identifying clinically relevant ADCP-related genes. This integration of bioinformatics prediction with functional validation establishes a paradigm for developing biologically grounded prognostic models that can guide both patient stratification and therapeutic target discovery in metastatic melanoma. (page 30, lines 630-636)

Reviewer #2 Comment 4: The explanation provided for omitting the heatmap of the 887 common DEGs due to “space limitations” is not fully convincing.

Response: We greatly appreciate your constructive feedback. We recognize the importance of the heatmap for the scientific rigor of the article. To address this concern, we have included a heatmap of the 887 common DEGs in Figure S2 in the Supplementary Materials. In addition, the expression levels of each gene have been summarized in Supplementary Table 1.

Reviewer #2 Comment 5: A schematic summary of the study workflow remains missing. Given the complexity of the multi-step bioinformatics and validation pipeline, a figure summarizing the analytical flow would greatly aid comprehension.

Response: We thank the reviewer for this suggestion. We have addressed this by:

1. Adding a workflow figure as Figure S1 in the Supplementary Materials.

2. Revising the Results section (page 14, lines 282-283) to reference this figure: " In this study, we implemented a multi‐step analytical pipeline that combined large‐scale bioinformatic screening with targeted experimental validation (Figure S1)."

3. Adding Figure S1 legend (page 50, Supplementary Figure Legends): "Figure S1. Workflow diagram of the study. The flowchart illustrates the systematic approach used in this research, including: (1) Data acquisition from TCGA and GEO databases; (2) Identification of differentially expressed genes between primary and metastatic melanoma; (3) Intersection with ADCP-related genes; (4) Feature selection using LASSO regression; (5) Construction of the six-gene prognostic model; (6) Validation in multiple cohorts; (7) Functional analyses including pathway enrichment, immune cell infiltration, and drug sensitivity prediction; (8) Experimental validation of DOCK10 through siRNA knockdown and functional assays."

Response to Reviewer #3

Reviewer #3 Comment: Authors have described the protocols of in vitro experiments in Methods section of the manuscript and that makes all of my comments addressed.

Response: We sincerely thank you for your positive feedback and for acknowledging that we have adequately addressed your concerns regarding the experimental protocols. Your previous suggestions to include detailed descriptions of the in vitro experiments have significantly improved the completeness and reproducibility of our work. We appreciate your time and effort in reviewing our manuscript.

---

## [Decision Letter · Decision Letter 2]

8 Sep 2025

PONE-D-25-08700R2Exploring the impact of Antibody-Dependent Cellular Phagocytosis-Related Genes on the prognosis of metastatic melanomaPLOS ONE

Dear Dr. Zhang,

Thank you for submitting your manuscript to PLOS ONE. After careful consideration, we feel that it has merit but does not fully meet PLOS ONE’s publication criteria as it currently stands. Therefore, we invite you to submit a revised version of the manuscript that addresses the points raised during the review process.

We look forward to receiving your revised manuscript.

Kind regards,

Xing-Xiong An, M.D.

Academic Editor

PLOS ONE

**Journal Requirements:**

Reviewers' comments:

Reviewer's Responses to Questions

**Comments to the Author**

1. If the authors have adequately addressed your comments raised in a previous round of review and you feel that this manuscript is now acceptable for publication, you may indicate that here to bypass the “Comments to the Author” section, enter your conflict of interest statement in the “Confidential to Editor” section, and submit your "Accept" recommendation.

Reviewer #2: (No Response)

2. Is the manuscript technically sound, and do the data support the conclusions?

Reviewer #2: Yes

3. Has the statistical analysis been performed appropriately and rigorously? 

Reviewer #2: Yes

4. Have the authors made all data underlying the findings in their manuscript fully available?

Reviewer #2: Yes

5. Is the manuscript presented in an intelligible fashion and written in standard English?

Reviewer #2: Yes

6. Review Comments to the Author

**Reviewer #2: ** We thank the authors for their careful revisions and constructive engagement with the review process. Only a few minor issues remain.

1. The abstract exceeds optimal length for PLOS ONE. Authors are recommended to suggest improving conciseness, grammar, and flow for clarity and readability. For instance,

Phrases such as “a nomogram was industrialized” (Line 36) should be revised (e.g., “constructed” or “developed”).

Line 51: “which is allied with poorer prognosis” should be rephrased as “associated with” for clarity.

Line 61: “prominence its potential utility” is unclear; consider revising to “highlighting its potential utility”.

Several molecular techniques (qRT-PCR, Western blotting, functional assays) are mentioned multiple times, causing repetition.

The background should focus on the scientific rationale and hypothesis, not techniques. A well-edited and concise abstract will better reflect the strength of the study.

2. The inclusion of heatmaps for the 887 shared DEGs across GSE7553 and GSE46517 is appreciated. However, the current visual presentation could be further improved:

The upper sample-type color bar is redundant, given that sample types are already represented in the lower legend. Its removal would streamline the figure.

The scale bar lacks appropriate labeling, and the normalization method is not indicated. Please clarify this either on the figure or in the legend.

Importantly, the heatmaps lack dendrograms, making it difficult to assess whether hierarchical clustering was performed on genes and/or samples. Including clustering branches would enhance interpretability and biological insight.

We recommend these final adjustments to enhance the clarity and rigor of the manuscript before acceptance.

7. PLOS authors have the option to publish the peer review history of their article (what does this mean? ). If published, this will include your full peer review and any attached files.

**Do you want your identity to be public for this peer review?** For information about this choice, including consent withdrawal, please see our Privacy Policy .

Reviewer #2: No

---

## [Author Response · Author response to Decision Letter 3]

15 Sep 2025

Reviewer #2 Comment 1:

The abstract exceeds optimal length for PLOS One. Authors are recommended to suggest improving conciseness, grammar, and flow for clarity and readability. For instance,

Phrases such as “a nomogram was industrialized” (Line 36) should be revised (e.g., “constructed” or “developed”).

Line 51: “which is allied with poorer prognosis” should be rephrased as “associated with” for clarity.

Line 61: “prominence its potential utility” is unclear; consider revising to “highlighting its potential utility”.

Several molecular techniques (qRT-PCR, Western blotting, functional assays) are mentioned multiple times, causing repetition.

The background should focus on the scientific rationale and hypothesis, not techniques. A well-edited and concise abstract will better reflect the strength of the study.

Response:

We sincerely thank the reviewer for the constructive comments and valuable suggestions that have greatly helped us improve the quality of our manuscript. The revisions made in response to these suggestions are detailed below:

1: We have condensed the abstract by removing redundant phrases and unnecessary details while retaining the core scientific findings and methods. The revised abstract is now under 300 words and more concise. We focused on the most relevant results and the significance of our findings for clinical decision-making in metastatic melanoma.

2: We have replaced "industrialized" with "developed" (Line 27) to align with standard academic writing, as suggested by the reviewer.

3: We have revised this phrase to "associated with poorer prognosis," (Line 37) which is more precise and academically appropriate.

4: The phrase has been revised to "highlighting its potential utility," (Line 41) improving clarity and flow, as recommended.

5: We have streamlined the mention of these techniques. The details are now included in the methods section only once, rather than repeating throughout the abstract. This reduces redundancy and keeps the focus on the key findings.

6: We have revised the background to focus more on the scientific rationale and research hypothesis, removing excess detail about the experimental techniques. This makes the abstract more focused on the core scientific questions and insights of the study.

Additional Changes Made:

Overall Clarity and Readability: We have revised the abstract to improve its flow and ensure it is easier to read and understand. Unnecessary technical jargon was reduced, and the overall language was simplified for better accessibility to a broad audience.

Thank you for your helpful suggestions! We believe the changes improve the clarity and impact of our abstract and manuscript.

Reviewer #2 Comment 2:

The inclusion of heatmaps for the 887 shared DEGs across GSE7553 and GSE46517 is appreciated. However, the current visual presentation could be further improved:

The upper sample-type color bar is redundant, given that sample types are already represented in the lower legend. Its removal would streamline the figure.

The scale bar lacks appropriate labeling, and the normalization method is not indicated. Please clarify this either on the figure or in the legend.

Importantly, the heatmaps lack dendrograms, making it difficult to assess whether hierarchical clustering was performed on genes and/or samples. Including clustering branches would enhance interpretability and biological insight.

We recommend these final adjustments to enhance the clarity and rigor of the manuscript before acceptance.

Response:

Thank you for the helpful suggestions regarding the heatmap presentation. We have made the following revisions:

1: Sample-type color bar: We removed the upper “sample-type” label as it duplicated the information already shown in the lower legend.

2: Scale bar and normalization: The heatmaps were generated with the R package pheatmap using scale = "row". This performs z-score normalization (value minus the row mean, divided by the row standard deviation). We have added this information next to the scale bar and in the figure legend.

3: Dendrograms: We now display dendrograms for genes in the revised figure, indicating hierarchical clustering performed after the row-wise z-score transformation. (Clustering in pheatmap was run with default settings.) The sample order is kept as in the experimental design and is stated in the legend.

We believe these changes improve clarity and interpretability, and we appreciate the reviewer’s constructive feedback.

---

## [Decision Letter · Decision Letter 3]

22 Sep 2025

Exploring the impact of Antibody-Dependent Cellular Phagocytosis-Related Genes on the prognosis of metastatic melanoma

PONE-D-25-08700R3

Dear Dr. Zhang,

We’re pleased to inform you that your manuscript has been judged scientifically suitable for publication and will be formally accepted for publication once it meets all outstanding technical requirements.

Kind regards,

Xing-Xiong An, M.D.

Academic Editor

PLOS ONE

Additional Editor Comments (optional):

Reviewers' comments:

Reviewer's Responses to Questions

**Comments to the Author**

1. If the authors have adequately addressed your comments raised in a previous round of review and you feel that this manuscript is now acceptable for publication, you may indicate that here to bypass the “Comments to the Author” section, enter your conflict of interest statement in the “Confidential to Editor” section, and submit your "Accept" recommendation.

Reviewer #2: All comments have been addressed

2. Is the manuscript technically sound, and do the data support the conclusions?

Reviewer #2: Yes

3. Has the statistical analysis been performed appropriately and rigorously? 

Reviewer #2: Yes

4. Have the authors made all data underlying the findings in their manuscript fully available?

Reviewer #2: Yes

5. Is the manuscript presented in an intelligible fashion and written in standard English?

Reviewer #2: Yes

6. Review Comments to the Author

Reviewer #2: I would like to thank the authors for their efforts in preparing this manuscript. Overall, the study has improved. I have no major concerns, and I believe the manuscript is suitable for publication in its current form.

7. PLOS authors have the option to publish the peer review history of their article (what does this mean? ). If published, this will include your full peer review and any attached files.

**Do you want your identity to be public for this peer review?** For information about this choice, including consent withdrawal, please see our Privacy Policy .

Reviewer #2: No

---

## [Editor Report · Acceptance letter]

PONE-D-25-08700R3

PLOS ONE

Dear Dr. Zhang,

I'm pleased to inform you that your manuscript has been deemed suitable for publication in PLOS ONE. Congratulations! Your manuscript is now being handed over to our production team.

Kind regards,

on behalf of

Dr. Xing-Xiong An

Academic Editor

PLOS ONE